# Molecular basis of PIP$_2$-dependent regulation of the Ca$^{2+}$-activated chloride channel TMEM16A

Son C. Le [1], Zhiguang Jia[2], Jianhan Chen [2,3] & Huanghe Yang [1,4]

The calcium-activated chloride channel (CaCC) TMEM16A plays crucial roles in regulating neuronal excitability, smooth muscle contraction, fluid secretion and gut motility. While opening of TMEM16A requires binding of intracellular Ca$^{2+}$, prolonged Ca$^{2+}$-dependent activation results in channel desensitization or rundown, the mechanism of which is unclear. Here we show that phosphatidylinositol (4,5)-bisphosphate (PIP$_2$) regulates TMEM16A channel activation and desensitization via binding to a putative binding site at the cytosolic interface of transmembrane segments (TMs) 3–5. We further demonstrate that the ion-conducting pore of TMEM16A is constituted of two functionally distinct modules: a Ca$^{2+}$-binding module formed by TMs 6–8 and a PIP$_2$-binding regulatory module formed by TMs 3–5, which mediate channel activation and desensitization, respectively. PIP$_2$ dissociation from the regulatory module results in ion-conducting pore collapse and subsequent channel desensitization. Our findings thus provide key insights into the mechanistic understanding of TMEM16 channel gating and lipid-dependent regulation.

[1] Department of Biochemistry, Duke University Medical Center, Durham, NC, USA. [2] Department of Chemistry, University of Massachusetts, Amherst, MA, USA. [3] Department of Biochemistry and Molecular Biology, University of Massachusetts, Amherst, MA, USA. [4] Department of Neurobiology, Duke University Medical Center, Durham, NC, USA. Correspondence and requests for materials should be addressed to H.Y. (email: huanghe.yang@duke.edu)

Ca$^{2+}$-activated chloride channels (CaCCs) are highly expressed in many cell types where they play important regulatory roles in neuronal firing, smooth muscle contraction, gastrointestinal motility, airway and exocrine gland secretion, and tumor cell proliferation and migration[1,2]. As the founding member of the transmembrane protein 16 (TMEM16) family, TMEM16A proteins are the *bona fide* pore-forming subunits of CaCC[3–5]. Aberrant upregulation of TMEM16A has been found in various pathological conditions, including head and neck squamous carcinomas, gastrointestinal stromal tumors, breast cancer, asthma and chronic obstructive pulmonary disease[2]. Owing to its importance in physiology as well as its therapeutic potential, there is an urgent need to understand the molecular underpinnings of TMEM16A–CaCC activation and regulation.

TMEM16A exhibits hallmarks of CaCCs, including Ca$^{2+}$- and voltage-dependent activation and nonselective anion permeability[3–5]. Whereas it requires membrane depolarization for activation under sub-micromolar cytosolic Ca$^{2+}$, TMEM16A becomes constitutively open under saturating Ca$^{2+}$ concentrations (above 1–2 µM). Structurally, TMEM16A channels adopt an architecture of a "double-barreled" homodimer with each monomer consisting of ten transmembrane (TM) helices with cytoplasmic N$^-$ and C-terminal domains[6–11] (Fig. 1a). TMs 3–8 of each monomer encompass a partially enclosed furrow that forms the putative ion permeation pathway[7–9,12,13]. Interestingly, the Ca$^{2+}$-binding site is formed by highly conserved acidic residues from TMs 6, 7, and 8[6,7,9,14,15]. This architecture suggests that Ca$^{2+}$ binding to the Ca$^{2+}$-sensing TMs 6–8 directly gates ion permeation in TMEM16A. Indeed, TM6 undergoes prominent conformational changes upon Ca$^{2+}$ binding to mediate channel activation[9]. The precise region responsible for TMEM16A's voltage-dependence still remains elusive although the first intracellular loop formed by the TM2–TM3 linker[16] and TM6[17] were implicated in voltage sensing.

In addition to Ca$^{2+}$- and voltage-dependent activation, time-dependent current decay or rundown is another hallmark of CaCCs, which has been observed in both endogenous CaCCs[18–20] and recombinantly expressed TMEM16A[7,21,22]. During prolonged stimulation by saturating Ca$^{2+}$, TMEM16A–CaCC current decreases over time, eventually leading to complete channel desensitization (Fig. 1b). This phenomenon has been alternatively referred to as "rundown", "desensitization" or "inactivation" in the literature; we will use desensitization and rundown interchangeably to describe the time-dependent TMEM16A current decay in this work. This desensitized state was likely captured in the recent Ca$^{2+}$-bound TMEM16A structures, where the putative ion permeation pore adopts a nonconductive collapsed configuration even though both Ca$^{2+}$ binding sites were fully occupied (Fig. 1a)[7,9]. While lack of permeant ions could contribute to this nonconducting conformation as described in the C-type inactivation observed in voltage-gated potassium channels[23], these observations are also indicative of a regulatory mechanism that controls channel opening of TMEM16A in addition to Ca$^{2+}$ and voltage. Indeed, Ca$^{2+}$-sensing calmodulin (CaM) was first suggested to regulate channel activation and inactivation of both TMEM16A and TMEM16B CaCCs[24–26]. However, this hypothesis has been challenged, as purified TMEM16A alone was sufficient to form Ca$^{2+}$-activated chloride channels[27]. Immunoprecipitation experiments also showed a lack of TMEM16A–CaM interaction, and exogenous manipulations of CaM exerted no effects on TMEM16A channel activation and permeability[14,21,27,28].

Phosphatidylinositol (4,5)-bisphosphate (or simply PIP$_2$), an important signaling lipid[29] that also acts as a key regulator of numerous ion channels[30,31], was recently proposed to regulate

TMEM16A channel gating[22,32–34]. This hypothesis was further bolstered by the observation that TMEM16F, a dual function TMEM16 ion channel and lipid scramblase[13,35], requires PIP$_2$ for its ion channel activity[36]. Despite these advances, the precise roles of PIP$_2$ in modulating TMEM16A channel gating are poorly defined, and the structural determinants that govern this lipid-dependent regulation of TMEM16A channel gating remain to be elucidated.

In this work, we combine mutagenesis, electrophysiology, and molecular modeling to demonstrate that PIP$_2$ facilitates TMEM16A's channel opening via binding to a putative binding site within TMs 3–5, herein referred to as the "PIP$_2$ binding module", that is distinct from the Ca$^{2+}$-binding site. Notably, the ion permeation pore of TMEM16A is comprised of residues from both the proposed "PIP$_2$ binding module" of TMs 3–5 and the "Ca$^{2+}$ binding module" of TMs 6–8, where each controls channel desensitization and Ca$^{2+}$-dependent channel activation, respectively. Binding of PIP$_2$ to the basic residues in the regulatory module stabilizes the open state of TMEM16A likely by impeding the gradual collapse of the ion conduction pore. Taken together, our results provide a structural framework to understand the interplay between PIP$_2$ regulation and Ca$^{2+}$ activation during channel gating in the TMEM16A–CaCC.

## Results

**PIP$_2$ stabilizes TMEM16A's open conformation under high Ca$^{2+}$.** Under saturating intracellular Ca$^{2+}$ concentration, the fully open TMEM16A channels undergo desensitization over time when recorded under inside-out configuration (Fig. 1b). TMEM16A currents measured at −80 mV and +80 mV decayed exponentially with identical rates to below detection limit within 4–5 min in 100 µM Ca$^{2+}$ (Fig. 1b, c). The time it takes for the channel to lose half of its initial current at −80 mV, denoted $t_{1/2}$, can be used to quantify the desensitization rate (Fig. 1c).

In addition to its important roles in a plethora of cellular signaling processes[29], PIP$_2$ is also known to regulate many ion channels and transporters[31,37]. Recent studies have revealed that PIP$_2$ modulates both endogenous and heterologously expressed TMEM16A under low intracellular Ca$^{2+}$ concentrations[32–34]. We therefore tested whether exogenously applied diC$_8$ PIP$_2$, a water-soluble short-chain analog of PIP$_2$, could alter TMEM16A desensitization under saturating intracellular Ca$^{2+}$. Consistent with a previous observation[22], we found that diC$_8$ PIP$_2$ substantially attenuates TMEM16A desensitization in a dose-dependent manner in the presence of 100 µM intracellular Ca$^{2+}$ (Fig. 1d–f). The effective diC$_8$ PIP$_2$ concentration required to maintain half of the initial current (EC$_{50}$) following 5 min of channel activation is ~3.95 µM, a value comparable to those reported in other well-characterized PIP$_2$-regulated ion channels such as KCNQ and K$_{ir}$ channels[38–40].

Both the full-length and the water-soluble short-chain diC$_8$ PIP$_2$ molecules can strongly impede TMEM16A desensitization. The effect of full-length PIP$_2$ on attenuating TMEM16A desensitization, however, is much more sustained than that of diC$_8$ PIP$_2$ upon removal from the cytosolic perfusion (Supplementary Fig. 1a–c). This phenomenon is likely due to the different acyl chain lengths of the PIP$_2$ molecules. The water-soluble short-chain diC$_8$ PIP$_2$ is more likely to be washed off the membrane than FL PIP$_2$, thus resulting in less sustained effect[41–43]. TMEM16A also exhibits phosphoinositide specificity as PI(3,4,5)P$_3$ significantly attenuates desensitization while PI(4)P exerts minimal effect on the channel, consistent with a recent study[34] (Supplementary Fig. 1d). We further found that 2 mM MgATP, but not Na$_2$ATP, significantly slowed down TMEM16A desensitization, presumably by promoting PIP$_2$ synthesis through

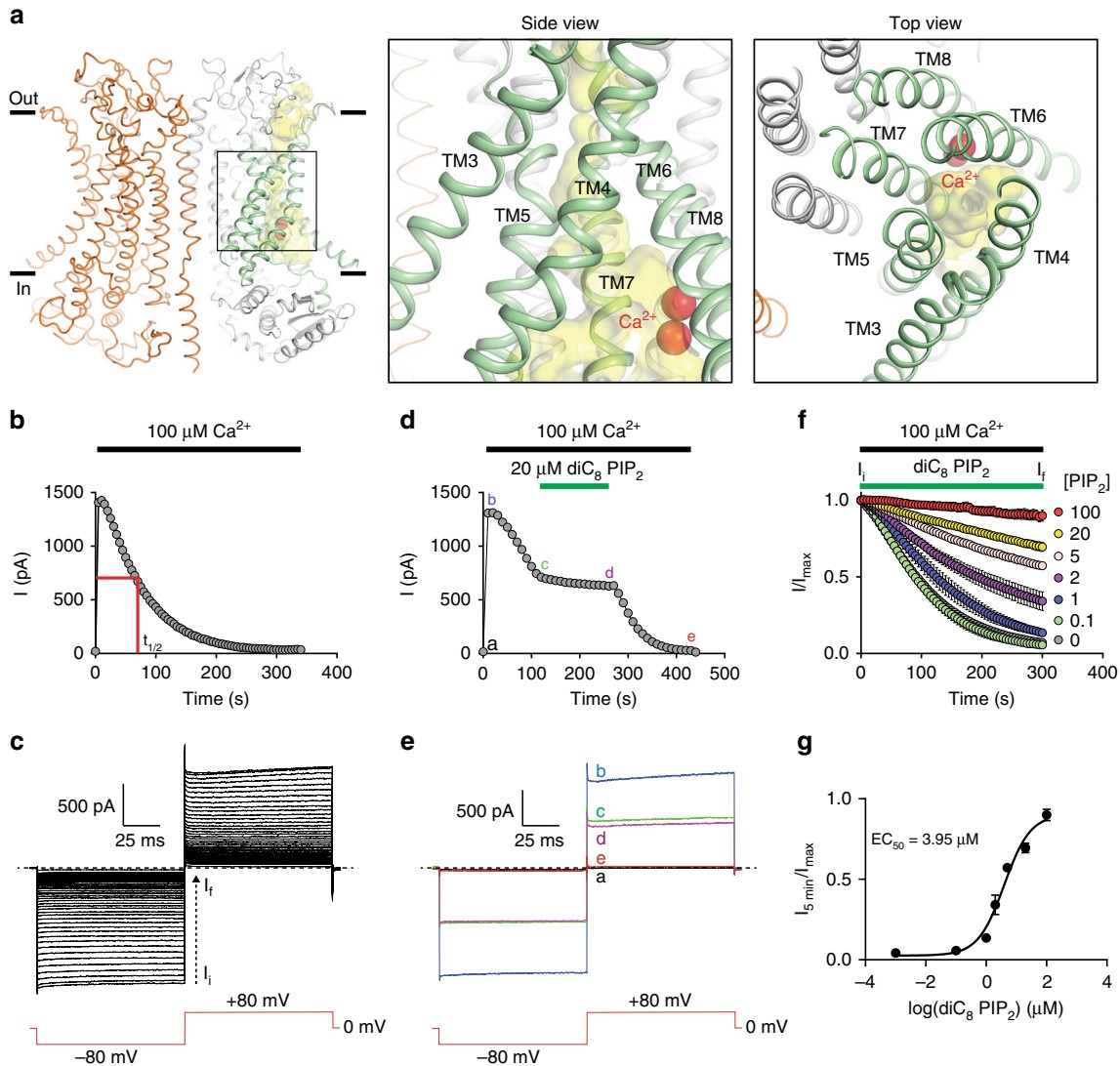

**Fig. 1** PIP$_2$ attenuates TMEM16A desensitization in saturating Ca$^{2+}$. **a** Overall architecture of the homodimeric Ca$^{2+}$-bound mouse TMEM16A with two monomers colored in orange and gray, respectively (PDB 5OYB). The yellow surface depicts the solvent-accessible volume of the putative permeation pore, which is in an apparently nonconductive state despite Ca$^{2+}$ binding (red spheres). Green helices encompass the permeation pathway. **b, c** Representative recording showing TMEM16A channel activity recorded using inside-out patch clamp under saturating 100 μM intracellular Ca$^{2+}$ (black bar). Currents were elicited by the voltage protocol shown in inset at a 5-s interval. $t_{1/2}$ denotes the half decay time (see "Methods"); $n = 23$ independent recordings. **d, e** Representative recording of TMEM16A under saturating 100 μM intracellular Ca$^{2+}$ (black bar) in the presence or absence of 20 μM diC$_8$ PIP$_2$ (green bar). Representative current traces at different time points in **d** are shown in **e**; $n = 8$ independent recordings. **f** diC$_8$ PIP$_2$ attenuates TMEM16A channel desensitization in a dose-dependent manner. I$_i$ and I$_f$ are the initial and final current amplitudes during the 5-min application of Ca$^{2+}$. **g** Dose-dependence of diC$_8$ PIP$_2$ in retaining TMEM16A channel activity. The sigmoidal curve represents fit to the Hill equation. The half-maximal concentration (EC$_{50}$) for diC$_8$ PIP$_2$ is 3.95 μM. $n = 4$ to 8 for each diC$_8$ PIP$_2$ concentration. Data are mean ± s.e.m. Source data are provided as a Source Data file

Mg$^{2+}$-dependent phosphoinositide kinases[44,45] (Supplementary Fig. 1e). These results further establish the critical role of PIP$_2$ in controlling TMEM16A desensitization.

We next examined whether exogenous PIP$_2$ can reactivate the completely desensitized TMEM16A channels after prolonged channel opening in saturating Ca$^{2+}$. Intriguingly, application of 20 μM diC$_8$ PIP$_2$ in the presence of 100 μM Ca$^{2+}$ yielded no detectable recovery of channel activity (Supplementary Fig. 2a, b). Interestingly, repeated treatments of the completely desensitized TMEM16A channels with full-length PIP$_2$ (10 μM) but not EGTA solution can partially restore TMEM16A activity (Supplementary Fig. 2c–e). These results suggest that under saturating Ca$^{2+}$, the fully-open TMEM16A likely enters an energetically stable desensitized state that is recalcitrant to reactivation by Ca$^{2+}$, voltage, and exogenous PIP$_2$.

**TMEM16A requires PIP$_2$ for opening under sub-micromolar Ca$^{2+}$.** TMEM16A can adopt multiple open states depending on intracellular Ca$^{2+}$[16,17]. As it exhibits voltage- and time-dependent activation under sub-micromolar Ca$^{2+}$ range[16], we next examined how PIP$_2$ may regulate TMEM16A in a more physiologically relevant Ca$^{2+}$ condition (0.25–0.50 μM). We found that PIP$_2$ depletion by poly-L-lysine (PLL, 100 μg/ml)[46] results in almost complete inhibition of the channel, which is rapidly reversed via exogenous diC$_8$ PIP$_2$ application (Fig. 2a–c) in a dose-dependent manner with an EC$_{50}$ of ~1.95 μM (Fig. 2d). This result suggests that TMEM16A likely displays an apparently high binding affinity for PIP$_2$, which is further substantiated by the observation that TMEM16A exhibits more sustained channel activity under sub-micromolar Ca$^{2+}$ (Fig. 2a and Supplementary Fig. 7a).

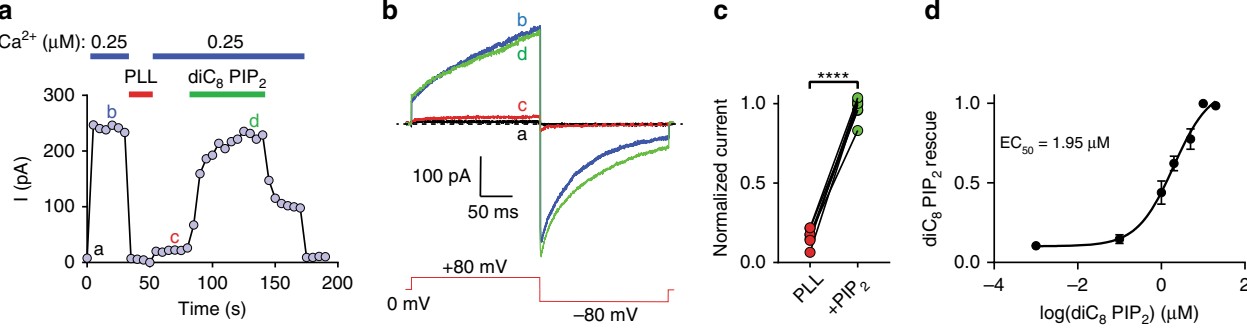

**Fig. 2** PIP$_2$ is required for TMEM16A's channel activation under sub-micromolar Ca$^{2+}$. **a, b** Representative time-course recording of TMEM16A under 0.25 µM intracellular Ca$^{2+}$ (blue bar). PIP$_2$ was depleted by poly-L-lysine (PLL, 100 µg/ml, red bar) and exogenous diC$_8$ PIP$_2$ (20 µM) was applied in the presence of 0.25 µM Ca$^{2+}$ (green bar). Representative raw current traces from different time points in **a** are shown in **b**. Repeated voltage protocol shown in inset was used to elicit TMEM16A current at a 5-s interval. **c** Normalized current responses after PLL treatment and during rescue by 20 µM diC$_8$ PIP$_2$. Two-tailed paired Student's $t$-test: $p$-value is < 0.0001. **d** Dose-response curve of diC$_8$ PIP$_2$-mediated rescue of TMEM16A after PLL-induced channel inhibition in the presence of 0.5 µM Ca$^{2+}$. The half-maximal rescue concentration (EC$_{50}$) of diC$_8$ PIP$_2$ is 1.95 µM. $n = 6$ to 12 for each diC$_8$ PIP$_2$ concentration. Data are mean ± s.e.m. Source data are provided as a Source Data file

To gain insights into the biophysical underpinnings of PIP$_2$-dependent regulation of TMEM16A under lower sub-micromolar Ca$^{2+}$, we applied stronger stimulations by increasing Ca$^{2+}$ concentration or membrane voltage following PIP$_2$ depletion-induced desensitization. Strikingly, stimulating the desensitized channels with higher Ca$^{2+}$ (Supplementary Fig. 3a) or higher membrane depolarization can restore channel activation (Supplementary Fig. 4a). We thus tested how PIP$_2$ depletion may alter TMEM16A's Ca$^{2+}$- and voltage-dependent gating by measuring the channel's Ca$^{2+}$ dose-dependent activation and voltage-dependent activation. We found that PLL treatment indeed reduces both the channel's apparent Ca$^{2+}$ sensitivity (Supplementary Fig. 3b–d) and voltage sensitivity (Supplementary Fig. 4b) as evidenced by the rightward shifts in the Ca$^{2+}$-dose response curve as well as the conductance–voltage ($G$–$V$) curve (Supplementary Fig. 4c, d). Taken together, we conclude that PIP$_2$ not only controls TMEM16A desensitization under saturating Ca$^{2+}$ but its binding is required for channel activation under sub-micromolar Ca$^{2+}$.

**Identification of putative PIP$_2$ binding residues in TMEM16A.** To uncover the molecular basis underlying regulation of TMEM16A's channel gating by PIP$_2$, we sought to identify its PIP$_2$ binding site(s) in TMEM16A. Previous functional and structural studies have unambiguously demonstrated that basic lysine and arginine residues at the membrane-cytosol interface usually form the PIP$_2$ binding sites via electrostatic interactions with the phosphate headgroups and phosphodiester linkage phosphate of PIP$_2$[31,47–49]. Neutralization of these residues should reduce the channel's PIP$_2$ binding affinity, thereby accelerating channel rundown. Guided by the atomic structures of a mouse TMEM16A[7,9], we conducted a systematic alanine mutagenesis scanning of all basic residues located approximately between the inner leaflet and the cytosol interface (Fig. 3a). By quantifying the time required for TMEM16A to lose half of its initial current ($t_{1/2}$) in saturating 100 µM Ca$^{2+}$ (Fig. 1c), we identified six basic residues whose neutralizations result in strongly enhanced desensitization, including R451 and K461 in TM2–3 linker, R482 in TM3, and K567, R575 and K579 in TMs 4 and 5 (Fig. 3a–c and Supplementary Figs. 5 and 6). When mapped to the TMEM16A structures[7,9], these residues are clustered near the cytosolic interface of TMs 3–5 (Fig. 3d). Notably, the R451A, K461A, R482A, K567A, R575A, and K579A mutations also strongly impair the effectiveness of exogenous PIP$_2$ in maintaining the open state of TMEM16A, consistent with their apparently

reduced PIP$_2$ binding affinities (Fig. 3c and Supplementary Fig. 6b–f). Under sub-micromolar Ca$^{2+}$, these mutations also pronouncedly accelerated spontaneous channel desensitization compared with WT TMEM16A (Supplementary Fig. 7). These results further support that the six basic amino acids from the alanine scanning may serve as putative PIP$_2$ binding residues.

To examine whether the observed mutational effects could result from the allosteric disruption of channel gating independent of PIP$_2$ binding, we examined the Ca$^{2+}$ sensitivity of these putative PIP$_2$ binding site mutations (Supplementary Fig. 8a–f). In sharp contrast to their pronounced effects on channel desensitization, mutations of the putative PIP$_2$ binding residues imposed no or minimal effects on the channel's apparent Ca$^{2+}$ sensitivity (Supplementary Fig. 8g, h). Based on the spatial proximity of these residues as well as their mutational effects on both TMEM16A desensitization and the effectiveness of PIP$_2$ to stabilize the channel's open state, we thus hypothesize that R451, K461, R482, K567, R575, and K579 identified from our mutagenesis screen constitute a PIP$_2$-binding site that regulates channel gating in TMEM16A.

**Validation of the putative PIP$_2$ binding site.** Molecular docking followed by atomistic simulations in explicit membrane and water (see "Methods") confirms that a full-length PIP$_2$ molecule can be stably accommodated in the putative binding site. The docked PIP$_2$ has its acyl chains positioned in the dimer cavity and its head group resided in the pocket concentrated with the positively charged residues identified in our mutagenesis scanning (Fig. 4a). The six putative PIP$_2$ binding residues likely play differential roles in PIP$_2$ binding. The sidechains of R451 and K567 directly coordinate PIP$_2$ (Fig. 4a), forming stable salt–bridge interactions with the 4′ or 5′-phosphate group of PIP$_2$ with probabilities of 0.78 ± 0.27 and 0.97 ± 0.02, respectively, during atomistic simulations (see "Methods" for details). Different from R451 and K567, R482 resides in an unresolved short peptide in the C-terminus of TM2–3 loop preceding TM3[7,9]. Based on our atomistic simulations, R482 has a relatively lower interaction probability of 0.60 ± 0.10 with PIP$_2$, likely owing to the flexibility of this region. Instead of interacting with the phosphate groups of PIP$_2$, R575 appears to form hydrogen bonds with the inositol ring and the phosphodiester linkage of PIP$_2$ with a high probability (0.86 ± 0.08) (Fig. 4a).

Despite its critical role in TMEM16A desensitization (Supplementary Fig. 6f), K579 does not appear to directly coordinate PIP$_2$ (Figs. 4a and 5a). This highly conserved residue in the

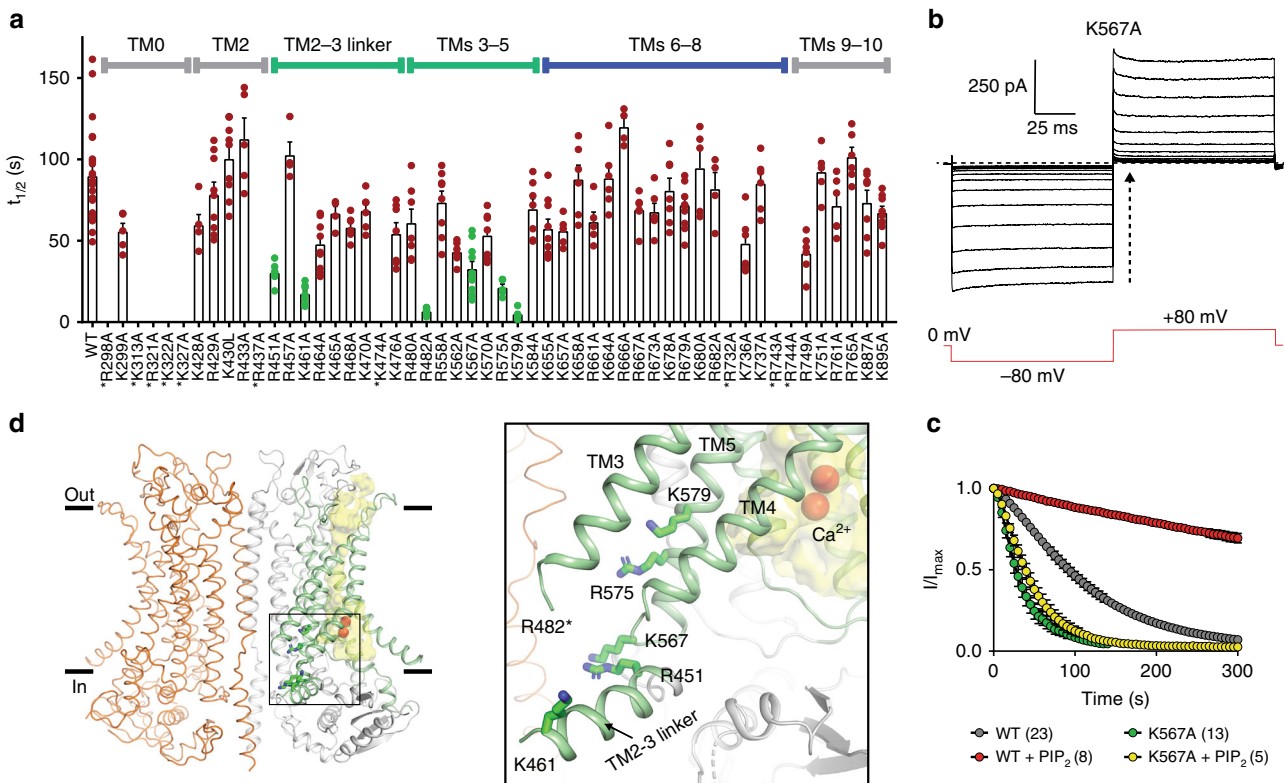

**Fig. 3** Systematic alanine scanning identified six basic residues critical for TMEM16A desensitization. **a** Summary of alanine mutagenesis screen of basic residues situated at the inner membrane and cytosol interface and their half-decay ($t_{1/2}$) values under 100 μM intracellular $Ca^{2+}$. The mutants exhibiting strongly accelerated desensitization rates are highlighted as green data points, including R451A, K461A, R482A, K567A, R575A, and K579A. Mutants that elicited negligible currents due to loss of function or surface expression are marked with an asterisk. $n$ ranges from 4 to 23 individual recordings for each mutant. **b** Representative current traces of K567A under saturating 100 μM $Ca^{2+}$. Currents were elicited by the voltage protocol shown in inset at a 5-s interval. **c** Average normalized currents of K567A in the presence of 100 μM $Ca^{2+}$ (green data points) or in the presence of 20 μM $diC_8$ $PIP_2$ and 100 μM $Ca^{2+}$ (yellow data points). WT TMEM16A's desensitization (gray) and its response to 20 μM $diC_8$ $PIP_2$ (red) are shown as controls. Numbers in parentheses denote the number of individual recordings. **d** Relative positions of the putative $PIP_2$ binding residues (green sticks). The ion permeation pathway is shown as yellow surface and the bound $Ca^{2+}$ ions are shown as red spheres. R482 is marked with an asterisk as it is not resolved in the cryo-EM structure (PDB 5OYB). Data are mean ± s.e.m. Source data are provided as a Source Data file

TMEM16 family (Supplementary Fig. 5), which is located in the middle of TM5, has a very low probability (0.05 ± 0.07) in interacting with the 1′-phosphate of $PIP_2$. Instead, K579 forms a stable salt-bridge interaction with the side chain of E564 in TM4 (Fig. 5a) as captured in the TMEM16A structures[7,9] It is thus likely that the profound effect of K579A on TMEM16A desensitization stems from the conformational alterations of the putative $PIP_2$ binding site due to the disruption of this salt bridge. Indeed, we found that alanine mutation of E564 also greatly accelerated TMEM16A desensitization (Supplementary Fig. 9a). Interestingly, this salt bridge is absolutely conserved in all TMEM16 proteins as well as the recently identified TMEM16-related mechano-/osmo-sensing OSCA/TMEM63 channels[50–52] (Supplementary Fig. 9b, c), further implicating the functional importance of this interaction in the TMEM16/TMEM63 superfamily of membrane proteins. Similar to K579, K461 residue in TM2–3 linker appears to be distally located from the docked $PIP_2$ (Fig. 5a) and thus is unlikely to coordinate $PIP_2$. We speculate that, analogous to K579A and E564A, K461A mutation may allosterically alter the conformation of the $PIP_2$ binding site, thereby indirectly disrupting $PIP_2$ binding.

To further validate the $PIP_2$ binding site, we tested whether full-length $PIP_2$ molecules could spontaneously bind to the putative $PIP_2$ binding site when randomly placed in the lipid bilayer using atomistic simulation. We observed that a full-length $PIP_2$ molecule placed near the binding pocket could enter the

putative binding pocket and rebind spontaneously within 50 ns (Fig. 4b and Supplementary Movie 1). The final configuration of $PIP_2$ from the rebinding simulation is strikingly similar to that obtained from the docking and equilibration simulations (Fig. 4b). However, no successful rebinding events were observed when $PIP_2$ was initially placed further away from the binding pocket and separated by one or more POPC lipids. This is likely due to the slow diffusion of phospholipid molecules within the lipid bilayer[53], as well as the slow binding of $PIP_2$ to the relatively narrow binding pocket. To circumvent this limitation, we simulated $PIP_2$-free $Ca^{2+}$-bound TMEM16A in a pure POPC membrane but with 20 copies of $PIP_2$ head group randomly placed in the bulk solvent. Remarkably, both binding pockets became occupied by the $PIP_2$ head groups within 200 ns of simulation. As illustrated in Supplementary Fig. 10 and Supplementary Movie 2, $PIP_2$ head groups diffused rapidly within the solution and reached the putative binding pocket from over 80 Å away, forming stable contacts with the identified putative $PIP_2$ binding residues. Collectively, our computational simulations and functional characterizations further validate the putative $PIP_2$ binding site as revealed by our alanine mutagenesis scanning.

**Distinct structural modules for $PIP_2$ and $Ca^{2+}$ binding**. The differential roles of the six putative $PIP_2$ binding residues as well as the functional importance of the highly conserved E564-K579

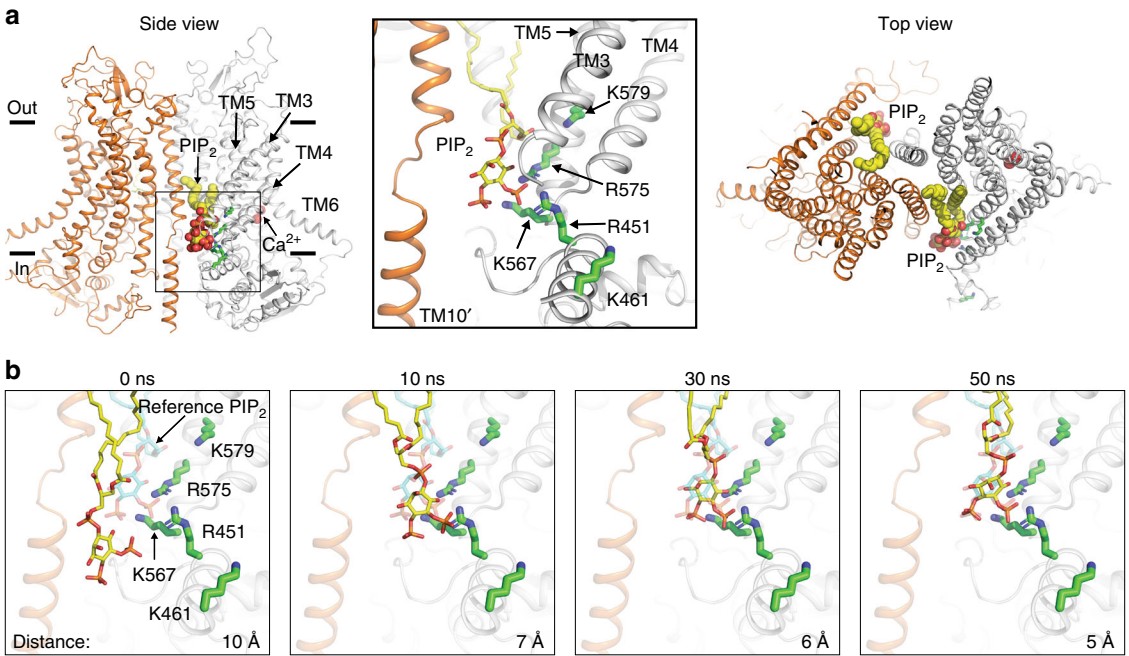

**Fig. 4** Structural model of $PIP_2$ bound to the putative binding site of TMEM16A. **a** Side and top view of a representative snapshot from a 400-ns atomistic simulation of the docked complex showing two well-equilibrated $PIP_2$ molecules (spheres) in the binding sites. Two TMEM16A monomers are shown as cartoon in orange and gray, respectively. Close-up view showing the potential interactions between $PIP_2$ and the putative $PIP_2$ binding residues (green sticks) is on the right. **b** Snapshots from the first 50 ns of a 200-ns atomistic simulation (see Supplementary Video 1) showing the process of a free $PIP_2$ molecule binding spontaneously to the same site in a configuration highly similar to the one obtained through docking followed by equilibration using atomistic simulations (shown as the "reference $PIP_2$" position in cyan sticks in all snapshots). The protein is shown as transparent cartoon, with four main $PIP_2$-contacting residues (R451, K567, R575, and K579) and the free $PIP_2$ molecule shown in green and yellow sticks, respectively. R482 is not shown as it is part of the highly flexible TM2–3 loop that was modeled into the structure. TM10 from the adjacent monomer is in orange. Note that $PIP_2$ remained in a similar bound configuration after 50 ns for the rest of the simulation. The distances between the centers of mass of $PIP_2$ and the key residues are also shown in the lower right of each panel

salt bridge imply that the structural integrity of the putative $PIP_2$ binding site is likely critical for TMEM16A desensitization. To test this possibility, we studied the mutational effects of a highly conserved proline (P566), which is located at TM4–5 linker (Fig. 5a). Because P566 may confer rigidity to TM4–5 linker, we hypothesize that increasing flexibility via an alanine mutation would perturb the conformation of the putative $PIP_2$ binding site, thereby accelerating desensitization. We observed that the P566A mutation profoundly enhances channel desensitization, consistent with the role of P566 in stabilizing the channel's open state (Fig. 5d). Remarkably, despite its effect on channel desensitization, P566A's apparent $Ca^{2+}$ sensitivity remains unaltered as compared with that of WT, suggesting that this mutation acts on TMEM16A channel gating independent of $Ca^{2+}$ (Fig. 5f, g). As a control, we also mutated P654, situated at the C-terminal cytoplasmic end of the $Ca^{2+}$-binding TM6, to alanine and examined its effects on TMEM16A's $Ca^{2+}$-dependent gating and desensitization (Fig. 5a). The P654A mutation reduces the channel's apparent $Ca^{2+}$ sensitivity (Fig. 5f, g), consistent with its role in TMEM16A's $Ca^{2+}$-dependent activation[9]. Remarkably, P654A's desensitization remains unaltered in spite of its reduced $Ca^{2+}$ sensitivity (Fig. 5d–g).

Similar to P566A, we found that mutations of two additional residues within the putative $PIP_2$ binding pocket, including D481, which is adjacent to the putative $PIP_2$ binding R482, and E564, which forms a salt–bridge interaction with K579 (Supplementary Fig. 9c), strongly enhance channel desensitization without altering their apparent $Ca^{2+}$ sensitivities (Fig. 5e–g and Supplementary Fig. 11a, b, g). By contrast, alanine mutations of key residues in the vicinity of the $Ca^{2+}$ binding sites recently

identified as important for channel activation, including G640 and Q645 in addition to P654[9,17], despite markedly altering their $Ca^{2+}$-dependent activation (Fig. 5a, g and Supplementary Fig. 11d–f, h), exert no discernable effects on channel desensitization (Fig. 5d, e).

As D481, E564, and P566 are located within the vicinity of the putative $PIP_2$ binding site whereas G640, Q645, and P654 are located on TM6 near the $Ca^{2+}$ binding site (Fig. 5a), their distinct functional roles are suggestive of a structural segregation of TMEM16A. The residues critical for channel desensitization are exclusively located in the cytosolic portion of TMs 3–5 facing the opposite side from the ion permeating "subunit cavity" (Figs. 3d and 5a). By contrast, the $Ca^{2+}$-binding acidic residues[14,15] and residues critical for $Ca^{2+}$-dependent gating are confined within TMs 6–8[9,17] (Figs. 3d and 5a). Intriguingly, our structural analyses reveal that TMs 3–8 of TMEM16A appear to form a pseudo twofold symmetric motif that encompasses the ion permeation pathway (Fig. 5a, b), also known as the "subunit cavity"[6]. Further confirming this observation, recent bioinformatics analyses also signaled at a potential structural repeat between TMs 3–5 and TMs 6–8 if only the transmembrane segments are taken into account[54]. These results thus provide evidence to support a two-module design of TMEM16A in which the ion permeation pathway is constituted of residues from the "$PIP_2$-binding regulatory module" formed by TMs 3–5 and the "$Ca^{2+}$-binding module" formed by TMs 6–8 (Fig. 5b, c).

**Ion permeation pore collapse causes TMEM16A desensitization.** The ion conduction pore of the recent $Ca^{2+}$-bound

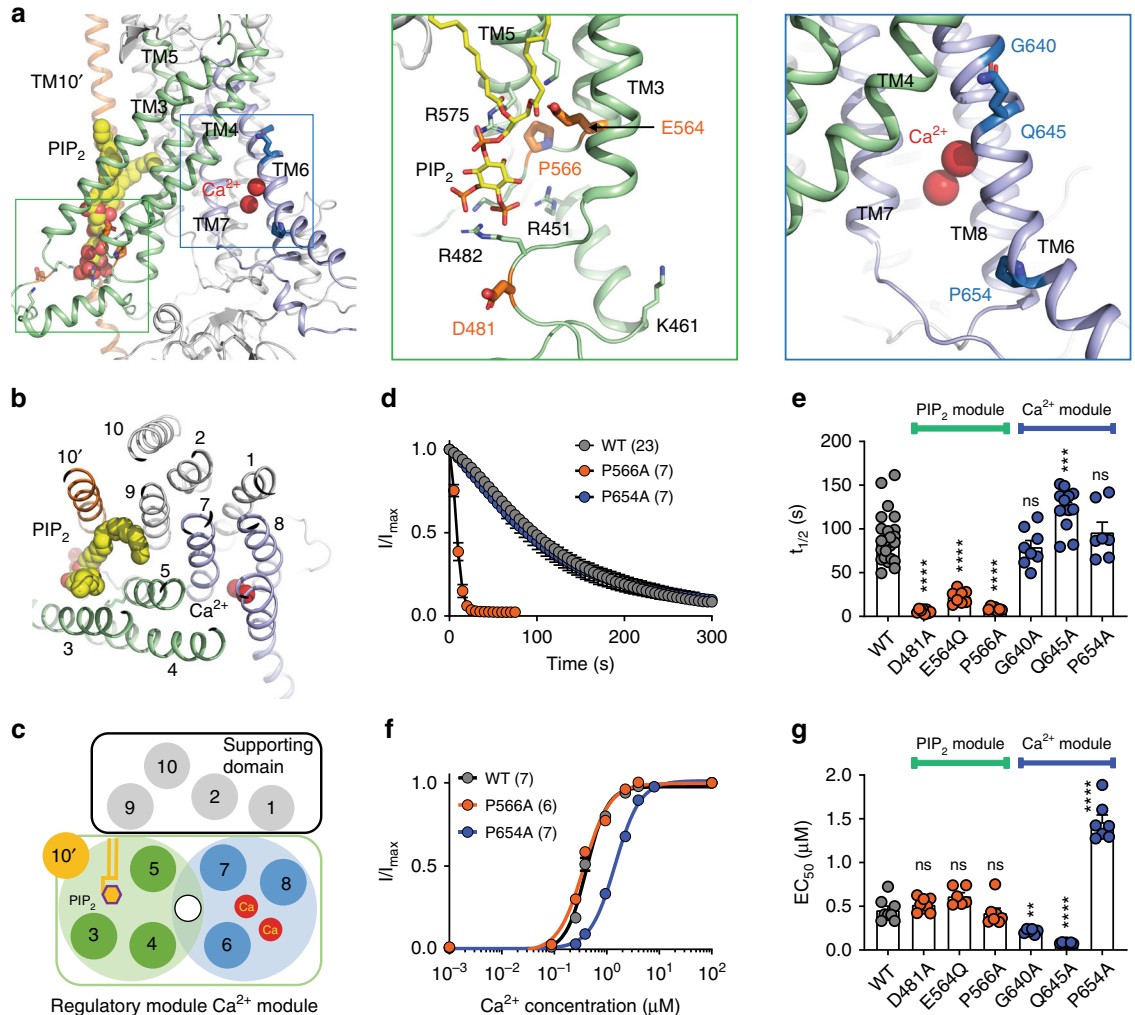

**Fig. 5** Two functionally and structurally distinct modules in TMEM16A. Side view (**a**) and top view (**b**) of a TMEM16A monomer with a docked PIP$_2$ (yellow and orange spheres) binding to TMs 3–5 (green) and Ca$^{2+}$ (red spheres) binding to TMs 6–8 (purple). The numbers mark TM segments from one subunit and TM10 from the second subunit (10'). D481 and R482 residues (marked with an asterisk) belong to the unresolved 21-amino acid loop that was modeled into the structure. **c** A schematic representation of the "two-module" design of TMEM16A ion permeation pore (top view). The PIP$_2$-binding regulatory module (TMs 3–5) and the Ca$^{2+}$-binding module (TMs 6–8) are shown in pale green and light purple, respectively. A supporting domain composing of TMs 1, 2, 9, and 10 is shown in gray. The anion permeation pathway (indicated by an unfilled circle) is formed between the Ca$^{2+}$ and PIP$_2$ modules. TM10 from the neighboring subunit (10') is colored in orange. **d** Average normalized currents of TMEM16 WT, P566A, and P654A under 100 μM Ca$^{2+}$. **e** Comparisons of the half decay time ($t_{1/2}$) of D481A, E564Q and P566A (thick orange sticks in **a**) and G640A, Q645A and P654A (thick blue sticks in **a**). **f** Ca$^{2+}$ dose-response curves of TMEM16A WT, P566A, and P564A. **g** Comparisons of the apparent Ca$^{2+}$ sensitivities of G640A, Q645A and P654A of the Ca$^{2+}$ module and D481A, E564Q, and P566A of the PIP$_2$ module. One-way ANOVA with Bonferroni's multiple comparisons test: $p$-values are <0.0001 for D481A, <0.0001 for E564Q, >0.0001 for P566A, >0.9999 for P654A, >0.9999 for G640A, 0.0006 for Q645A in **e**; >0.9999 for D481A, 0.1342 for E564Q, >0.9999 for P566A, <0.0001 for P654A, 0.0043 for G640A, <0.0001 for Q645A in **g**. Numbers in parentheses denote the number of individual recordings. Data are mean ± s.e.m. ns, not significant. Source data are provided as a Source Data file

TMEM16A structures appears to adopt an apparently collapsed conformation[7,9] (Fig. 1a). As the Ca$^{2+}$ binding sites are fully occupied whereas PIP$_2$ was not present in these structures, we hypothesize that TMEM16A undergoes desensitization via constricting its ion permeation pore during prolonged activation under saturating Ca$^{2+}$ in addition to PIP$_2$ dissociation. We thus employed three different approaches in which we perturbed the ion permeation pore and determined their effects on TMEM16A desensitization.

First, we examined the effect of SCN$^-$ on TMEM16A desensitization. SCN$^-$ has two distinct permeation characteristics in CaCCs[55]: SCN$^-$ is a larger anion with higher permeability than Cl$^-$ due to its lower hydration energy; SCN$^-$ has stronger interaction with the pore, which enhances its occupancy with the

pore. When replacing the majority of Cl$^-$ from both the intracellular and extracellular solutions with SCN$^-$, we found that TMEM16A undergoes significantly slower desensitization (Fig. 6a). This result is suggestive of a possibility that the stronger interaction between the large anion SCN$^-$ and the ion permeation pore could partially prevent pore collapse, thereby slowing down channel desensitization. Similarly, we observed that Br$^-$, which is more permeable than Cl$^{-12}$, also considerably reduced TMEM16A desensitization albeit less pronounced than SCN$^-$, presumably owing to Br$^-$'s smaller size compared with SCN$^-$(Supplementary Fig. 12). By contrast, replacing the majority of the permeating Cl$^-$ with the impermeant methanesulfonate (MES$^-$)[56] did not alter TMEM16A desensitization. As diC$_8$ PIP$_2$ dose-dependently attenuates TMEM16A

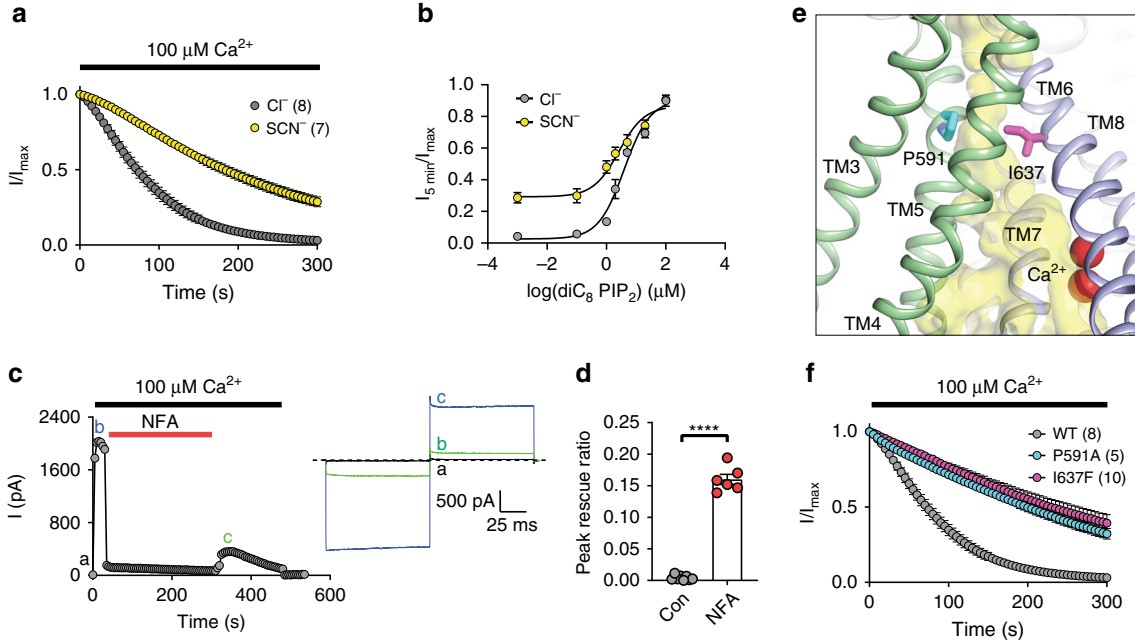

**Fig. 6** Disrupting the ion permeation pore alters TMEM16A desensitization. **a** Average normalized currents of TMEM16A measured in symmetric $Cl^-$ or $SCN^-$ condition under 100 µM $Ca^{2+}$. Intracellular and extracellular solutions are symmetric and contain 140 mM $Cl^-$($Cl^-$) or 112 mM $SCN^-$/28 mM $Cl^-$ ($SCN^-$). Numbers in parentheses denote the number of individual recordings. **b** Effects of $SCN^-$ on $diC_8$ $PIP_2$ dose-response of TMEM16A. $n = 6$–12 for each $diC_8$ $PIP_2$ concentration. **c, d** Representative recording of TMEM16A current in the presence of 300 µM niflunic acid (NFA) and 100 µM $Ca^{2+}$ from intracellular side. Current traces at different time points are shown on the right. Quantifications of peak residual current at 5 min after NFA washout or in control recordings are shown in **d**. Two-tailed unpaired Student's t-test: p-value is <0.0001. **e** Side view of mTMEM16A monomer showing the positions of two pore-lining residues P591 (cyan sticks) and I637 (magenta sticks). **f** Mutational effects of P591A and I637F on TMEM16A desensitization. Numbers in parentheses denote the number of individual recordings. Data are mean ± s.e.m. Source data are provided as a Source Data file

desensitization, likely by stabilizing the open conformation of the pore (Fig. 1f, g), we hypothesize that $SCN^-$ will further enhance the effects of $diC_8$ $PIP_2$. We thus tested the dose-dependent effects of $diC_8$ $PIP_2$ on TMEM16A desensitization and found that $SCN^-$ indeed enhances the ability of $diC_8$ $PIP_2$ to maintain TMEM16A channel activity as observed by the leftward shift in the $diC_8$ $PIP_2$ $EC_{50}$ curve (Fig. 6b). These results suggest that larger and more permeable anions likely interfere with the ion permeation pore to attenuate TMEM16A desensitization.

Second, we examined the effect of intracellular niflunic acid (NFA), a classical CaCC blocker[57], on TMEM16A desensitization. While the exact NFA binding site in TMEM16A has not been established, the ability of NFA to block CaCCs from both sides of the membrane[13,58,59] suggests that NFA likely interacts with the ion permeation pore. NFA is known to block TMEM16A without modifying its $Ca^{2+}$-dependent channel gating[57]. By blocking TMEM16A's channel activity with saturating 300 µM NFA in the presence of 100 µM $Ca^{2+}$ for 5 min, we detected a significant amount (~15%) of functional TMEM16A channels following NFA washout (Fig. 6c, d). This residual channel activity stands in contrast to the nearly complete loss of channel activity without NFA under saturating $Ca^{2+}$ (see Fig. 1b, c). To eliminate the possibility that the observed residual TMEM16A activity is due to the lack of ion permeation during NFA application, we replaced most of intracellular $Cl^-$ with the impermeant $MES^-$. Distinct from NFA blockade, $MES^-$ did not preserve functional TMEM16A channels after prolonged activation by saturating $Ca^{2+}$ (Supplementary Fig. 13). Thus, NFA blockade of the pore but not the lack of ion permeation is responsible for attenuating TMEM16A desensitization. It is tempting to postulate that NFA might act on TMEM16A via a "foot-in-the-door" mechanism[60] to stabilize the ion conduction pore.

Lastly, we directly altered the pore properties by mutating P591 in TM5 and I637 in TM6, two pore-lining residues that are located in the middle of the membrane and are distal from the $Ca^{2+}$ binding site and our proposed putative $PIP_2$ binding site (Fig. 6e). Both P591A and I637F mutations largely reduce TMEM16A desensitization under saturating $Ca^{2+}$ (Fig. 6f and Supplementary Fig. 14a, b). Consistent with its location within the permeation pathway and its role as an important activation gate residue[17,61], I637F alters TMEM16A's ion selectivity in addition to enhancing its apparent $Ca^{2+}$ sensitivity (Supplementary Fig. 14c, e, f, h–j). P591A, on the other hand, displays no detectable change in its ion selectivity but has an enhanced $Ca^{2+}$ sensitivity (Supplementary Fig. 14d, f, g, i, j).

Taken together, these three lines of functional evidence from experiments testing the effects of altering the ion permeation pore corroborate our hypothesis that pore collapse is likely the structural underpinning of TMEM16A desensitization under saturating $Ca^{2+}$ (Figs. 1a and 7).

## Discussion

Combining electrophysiology with intensive structure-guided mutagenesis, we identify the molecular mechanism underlying the multifaceted role of $PIP_2$ in modulating TMEM16A. Strengthened by our computational studies, our studies unravel a regulatory mechanism of TMEM16A channel gating by which $PIP_2$ binds to a putative binding site located near the cytosolic interface of TMs 3–5 to stabilize the open state of the permeation pathway of TMEM16A.

We herein propose a simplified "two-module" model of the ion permeation pore to explain the mechanism of how $PIP_2$ may regulate TMEM16A channel gating in a structural context (Figs. 5c and 7). We posit that binding of $Ca^{2+}$ to the

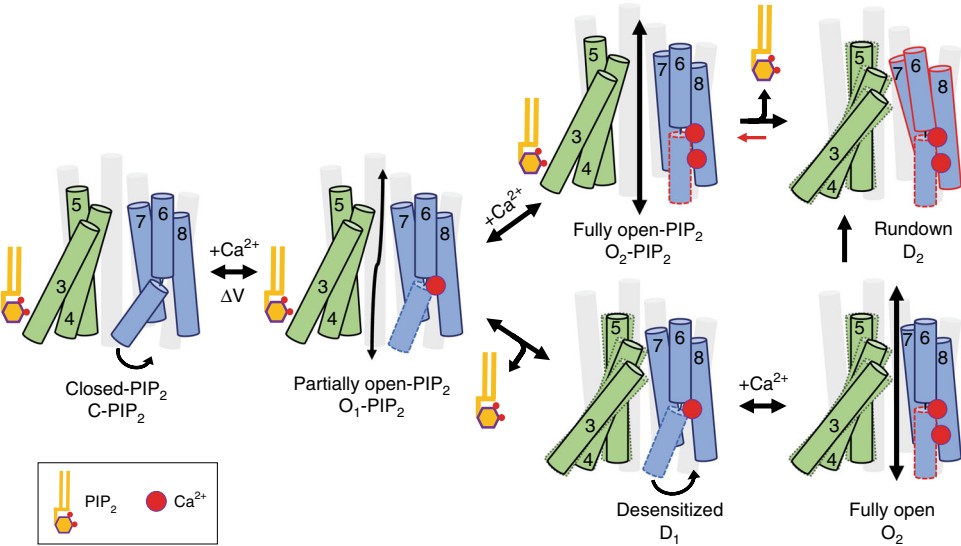

**Fig. 7** Proposed mechanistic model of PIP$_2$-dependent modulation of TMEM16A gating. The ion permeation pore is formed by two distinct structural modules that bind to PIP$_2$ (TMs 3–5, green) and Ca$^{2+}$ (TMs 6–8, blue), respectively. The remaining TMs (1, 2, 9, and 10) are shown in light gray. Ca$^{2+}$ binding to the Ca$^{2+}$ module of a closed channel (C-PIP$_2$) opens the ion permeation pore, whereas PIP$_2$ binding to the regulatory module stabilizes this open conformation (O$_1$-PIP$_2$). Depletion of PIP$_2$ from the O$_1$-PIP$_2$ state reversibly desensitizes the channel (D$_1$), which can be opened by higher Ca$^{2+}$ (O$_2$). Prolonged channel opening under saturating Ca$^{2+}$ (O$_2$-PIP$_2$) induces both PIP$_2$ dissociation from the regulatory module that subsequently results in an energetically stable closed state (D$_2$) that is resistant to reactivation by exogenous PIP$_2$

"Ca$^{2+}$-binding module" of TMs 6–8 provides the primary impetus to open the channel pore, mainly through, but not limited to, the conformational changes of TM6[9,17] (Fig. 7). Binding of PIP$_2$ to the "regulatory module" of TMs 3–5 enables the stabilization of the open pore conformation of TMEM16A in a wide range of Ca$^{2+}$. On the one hand, sub-micromolar Ca$^{2+}$ and voltage promote TMEM16A to enter a partial open state with bound endogenous PIP$_2$ (O$_1$-PIP$_2$). Depletion of PIP$_2$ rapidly converts the O$_1$-PIP$_2$ state to a transient desensitized state (D$_1$) that can either be reversed back to the O$_1$-PIP$_2$ by exogenous PIP$_2$ or be overcome by more Ca$^{2+}$ or depolarizing voltage to adopt a fully open state (O$_2$) (Fig. 7). On the other hand, following prolonged Ca$^{2+}$ activation, dissociation of PIP$_2$ from the fully open channel plays a pivotal role in the destabilization of the open pore conformation that is accompanied by a slow transition of the channel into an energetically stable desensitized (D$_2$) state, which is resistant to reactivation by Ca$^{2+}$, voltage, and exogenous PIP$_2$. Our functional characterizations using the large anion SCN$^-$, the CaCC blocker NFA, and pore mutations suggest that a "pore collapse" mechanism may underlie this high Ca$^{2+}$-induced desensitization in TMEM16A. This collapsed, nonconducting D$_2$ state was likely captured in the recent Ca$^{2+}$-bound TMEM16A structures in which the channel's steric gate adopts an apparently constricted conformation that precludes ion permeation[7,9]. Taken together, our findings illuminate a critical structural framework for the mechanistic understanding of lipid-dependent modulation of channel gating in TMEM16 ion channels.

Notably, the ion permeation pathway of TMEM16A is established by residues from both the Ca$^{2+}$-binding module and the PIP$_2$ regulatory module, both of which appear to form a structural repeat[54] (Fig. 5a–c). This unique structural feature implies that Ca$^{2+}$ and PIP$_2$, by binding to their modules, exert direct effects on the channel pore, thereby controlling TMEM16A gating and desensitization, respectively. In light of their evolutionarily conserved structures[7,9,62], we anticipate that this principle of dual regulation of TMEM16A by Ca$^{2+}$ and PIP$_2$ could potentially be generalized to other members of the TMEM16 family, including the Ca$^{2+}$-dependent scramblase TMEM16F[36]. Interestingly, the

salt bridge between E564 and K579 within our proposed regulatory PIP$_2$ module of TMEM16A is absolutely conserved among all TMEM16 members as well as the newly discovered mechano- or osmo-sensitive OSCA[50,51] and TMEM63 channels (Supplementary Fig. 9b, c). Given the high similarity in their transmembrane regions, we speculate that these evolutionarily- and structurally-related transport proteins might utilize similar conformational transitions during gating, even though they sense different physiochemical stimuli and permeate structurally distinct substrates.

In both nonexcitable and excitable cells, TMEM16A–CaCC can be activated by intracellular Ca$^{2+}$ following receptor-mediated activation of phospholipase C (PLC). Our current study together with previous reports[22,32] strongly support the notion that PLC-induced PIP$_2$ hydrolysis not only leads to TMEM16A channel activation via the IP$_3$ signaling pathway, but concomitantly the subsequent reduction in plasma membrane PIP$_2$ concentration could provide a negative feedback to turn off TMEM16A. We believe that this built-in PIP$_2$-dependent "brake" allows fine-tuning of TMEM16A–CaCC's activities during Ca$^{2+}$ signaling. Further studies are needed to interrogate the physiological relationship between TMEM16A and PIP$_2$ and whether this autonomous regulatory mechanism exists in other TMEM16 proteins.

## Methods
**Constructs and mutagenesis**. Mouse TMEM16A[7,12,14] (mTMEM16A, NCBI: NP_001229278, lacking isoform c of EAVK segment) tagged with a C-terminal eGFP or mCherry (in an N1 or a PUNIV vector) was used for all experiments. Single point mutations were generated using QuikChange site-directed mutagenesis kit (Agilent). Primers for PCR were purchased from IDT DNA Technologies (Supplementary Table 1). All mutant constructs were subsequently confirmed by Sanger sequencing from Genewiz.

**Cell culture and transfection**. HEK293T cells, authenticated and tested negative for mycoplasma by Duke Cell Culture Facility (ATCC CRL-11268), were grown in DMEM supplemented with 10% FBS (Sigma), 1% penicillin and streptomycin (Gibco), and supplied with 5% CO$_2$ at 37 °C. Cells grown on poly-L-lysine (PLL, Sigma) and laminin (Sigma) coated coverslips placed in a 24-well plate (Eppendorf) reaching 40–60% confluency were transiently transfected using X-tremeGENE 9 DNA transfection reagent (Sigma) at a 1:2 (m:v) DNA:transfection reagent ratio. Typically, 200 ng of TMEM16A DNA was used for each well of a 24-well plate to yield sufficient channel expression for recordings.

**Electrophysiology**. All electrophysiology recordings were carried out at room temperature within 24–48 h after transfection. Inside-out patch clamp recordings were performed on cells expressing fluorescently-tagged TMEM16A channels (mCherry or eGFP) with glass electrodes pulled from borosilicate capillaries (Sutter Instruments). Pipettes were fire-polished using a microforge (Narishge) and had a resistance of 2–3 MΩ. After formation of gigaohm seal, inside-out patches were excised from cells expressing TMEM16A constructs of interest. The pipette solution (external) contained (in mM): 140 NaCl, 5 EGTA, 10 HEPES, 2 MgCl₂, adjusted to pH 7.3 (NaOH), and the bath solution (internal) contained 140 mM NaCl, 10 mM HEPES, adjusted to pH 7.3 (NaOH). Internal free $Ca^{2+}$ solutions (<100 μM) were made by adding $CaCl_2$ at concentrations calculated using WEB-MAXC online software (http://maxchelator.stanford.edu/webmaxc/webmaxcE.htm) to an EGTA-buffered solution containing (in mM): 140 NaCl and 10 HEPES, 5 EGTA, adjusted to pH 7.3 (NaOH). Free $Ca^{2+}$ concentrations of EGTA-buffered $Ca^{2+}$ solutions were subsequently measured by a spectrometer using the ratio-metric $Ca^{2+}$ dye Fura-2 (ATT Bioquest), and using a $Ca^{2+}$ standard curve generated from a $Ca^{2+}$ calibration buffer kit (Biotum). Solutions containing 100 μM $Ca^{2+}$ or more were made by directly adding $CaCl_2$ into a solution containing (in mM): 140 NaCl and 10 HEPES, pH 7.3 (NaOH).

Application of the intracellular solution to excised inside-out patches was performed using a pressurized perfusion apparatus (ALA-VM8, ALA Scientific Instruments) in which the perfusion outlet was directly positioned in front of the pipette tip containing the excised membrane patch. TMEM16A channels were activated by focal perfusion of solutions containing the desired $Ca^{2+}$ concentrations to the cytoplasmic side of the excised patches.

For the rundown protocol in saturating $Ca^{2+}$, current responses were elicited by voltage steps of −80 and +80 mV lasting 100 ms each at an inter-sweep interval of 5 s, and the membrane was held at 0 mV. Steady state currents measured at −80 mV or +80 mV voltage steps were used for analysis.

For $Ca^{2+}$ concentration-dependent recordings, the membrane was held at +60 mV, and TMEM16A's outward currents were elicited by perfusion of EGTA-buffered $Ca^{2+}$ solutions of various free $Ca^{2+}$ concentrations. Quasi-steady state currents were measured for each $Ca^{2+}$ application and normalized to the peak current elicited by 100 μM $Ca^{2+}$ to construct the dose-dependent curve.

For phosphoinositide experiments, diC₈ PI(4)P, diC₈ PI(4,5)P₂, diC₈ PI(3,4,5)P₃ (Echelon Biosciences or Cayman Chemical Company), and full-length brain PI (4,5)P₂ (Avanti) were first dissolved in water to generate stock solutions of 5 and 0.5 mM for diC₈ lipids and full-length lipids, respectively. All lipid stock solutions were kept at −80 °C for long-term storage. On the day of recordings, phosphoinositide lipids were diluted in recording solutions containing desired free $Ca^{2+}$ concentrations, sonicated on ice for 5–10 min, and loaded into the pressurized perfusion chamber. All phosphoinositide-containing solutions were used within 2 h.

For PIP₂-mediated rescue experiments in sub-micromolar $Ca^{2+}$, inside-out patches from cells expressing mTMEM16A were perfused with the desired sub-micromolar $Ca^{2+}$ and +80 and −80 mV steps lasting 200 ms each were applied to elicit outward and inward currents, respectively. The inter-sweep interval was 5 s and the membrane was held at 0 mV. Channel activity was measured for 20–30 s to establish a baseline. Membrane PIP₂ depletion was then induced by ~15–20-s application of $Ca^{2+}$-free (5 mM EGTA) solution containing 100 μg/ml PLL or a solution with sub-micromolar $Ca^{2+}$ solution containing 100 μg/ml PLL to induce desensitization. diC₈ PIP₂-containing solutions with the desired sub-micromolar $Ca^{2+}$ were then applied to rescue channel activity. Steady state currents at the end of each +80-mV depolarization step were used for analyses.

To probe the effects of PIP₂ depletion on the voltage-dependent activation of TMEM16A, a two-step procedure was used to construct G–V relationships. First, using inside-out patches, we obtained maximal channel activation of TMEM16A (under 100 μM $Ca^{2+}$) using a voltage step protocol consisting of +120 mV depolarization step and −60 mV repolarization step, the tail current from which was used to calculate the $G_{max}$. To assess the effects of PLL, the membrane patch was first treated with PLL (100 μg/ml) for 15–17 s before 100 μM $Ca^{2+}$ was applied to obtain the $G_{max}$. Subsequently, the voltage-dependent activation of TMEM16A was quickly obtained by applying sub-micromolar $Ca^{2+}$ concentrations of 0.255 and 0.387 μM $Ca^{2+}$ using an I–V protocol in which the membrane was held at −60 mV, and voltage steps from −120 to +120 mV were applied to elicit channel opening. In both control and PLL-treated conditions, the tail currents measured at the −60-mV repolarization step in response to various voltages under 0.255 or 0.387 μM $Ca^{2+}$ were normalized to the tail current measured at −60 mV under 100 μM $Ca^{2+}$ to obtain the $G/G_{max}$–V curves.

For experiments with SCN⁻, both the extracellular (pipette) and intracellular (perfusion) solutions were symmetric in permeant ions (Cl⁻ and SCN⁻) and contained either 140 mM Cl⁻ (control) or 28 mM Cl⁻ and 112 mM SCN⁻(SCN⁻). All intracellular solutions also contained 100 μM $Ca^{2+}$. In experiments testing the effects of SCN⁻ on TMEM16A's PIP₂ sensitivity, internal and external solutions were symmetric and contained 28 mM Cl⁻ and 112 mM SCN⁻, and the internal solution contained 100 μM $Ca^{2+}$ with various concentrations of diC₈ PI(4,5)P₂ (0.1, 1, 2, 5, 20, and 100 μM). The normalized currents measured at 5 min were used to construct the dose-dependent curves showing effects of SCN⁻ on TMEM16A's PIP₂ sensitivity.

For reversal potential measurements, inside-out configuration was used for rapid exchange of the intracellular solution (bath solution). First, TMEM16A

channels were activated in which the pipette solution and perfusion solution (symmetric 140 mM Cl⁻) both contained (in mM): 140 NaCl, 5 EGTA, 10 HEPES, adjusted to pH 7.3 (NaOH) and osmolarity of ~300 (D-mannitol). The intracellular solution (perfusion) also contained 100 μM $Ca^{2+}$ to activate TMEM16A, and inward and outward currents were elicited by a repeated ramp protocol from −100 to +100 mV. Intracellular solution was then replaced by perfusion with a solution (low 14 mM NaCl) containing (in mM): 14 NaCl, 5 EGTA, 10 HEPES, adjusted to pH 7.3 (NaOH), 100 μM $Ca^{2+}$ and osmolarity of ~300 (adjusted with D-mannitol). The reversal potential ($E_{rev}$) was determined as the membrane potential at which the current was zero. The shift in $E_{rev}$ was calculated by subtracting the $E_{rev}$ of intracellular 14 mM NaCl from the $E_{rev}$ measured in symmetric 140 mM NaCl.

All electrophysiology recordings were low-pass filtered at 5 kHz (Axopatch 200B) and digitally sampled at 10 kHz (Axon Digidata 1550 A) and digitized by Clampex 10 (Molecular Devices).

**Data analysis**. All offline data analysis was performed using Clampfit, Microsoft Excel, and MATLAB (MathWorks). To quantify the extent of rundown in saturating $Ca^{2+}$, 100 μM $Ca^{2+}$-elicited currents measured at −80 or +80 mV were normalized to the peak current amplitude, and the half-decay time ($t_{1/2}$) was calculated by fitting the channel current decay with a curve using a custom-written MATLAB script.

For quantification of $Ca^{2+}$ dose-dependent concentrations ($EC_{50}$), normalized $Ca^{2+}$-induced currents were fit into a nonlinear regression curve fit of four parameters with the equation:

$$\frac{I}{I_{max}} = \frac{1}{1 + \left(\frac{[EC_{50}]}{[Ca^{2+}]}\right)^H} \tag{1}$$

where $I/I_{max}$ denotes current normalized to the max current elicited by 100 μM $Ca^{2+}$, $[Ca^{2+}]$ denotes free $Ca^{2+}$ concentration, H denotes Hill coefficient, and $EC_{50}$ denotes the half-maximal activation concentration of $Ca^{2+}$.

The permeability ratio $P_{Cl}/P_{Na}$ was calculated using the Goldman–Hodgkin–Katz equation:

$$E_{rev} = \frac{RT}{F} \ln \frac{P_{Na}[Na]_o + P_{Cl}[Cl]_i}{P_{Na}[Na]_i + P_{Cl}[Cl]_o} \tag{2}$$

where $\Delta E_{rev}$ is the measured shift in reversal potential (in mV); $P_{Na}$ and $P_{Cl}$ are the relative permeabilities of TMEM16A toward Na⁺ and Cl⁻, respectively; $[Na]_o$ and $[Na]_i$ are external and internal sodium concentrations, respectively; $[Cl]_i$ and $[Cl]_o$ are external and internal chloride concentrations, respectively; F is the Faraday's constant (96,485 C mol⁻¹), R is the gas constant (8.314 J mol⁻¹), and $T$ is the absolute temperature (298.15 K at 25 °C).

In current rescue experiments by diC₈ PIP₂ under sub-micromolar $Ca^{2+}$, PIP₂-mediated current rescue ratio was calculated by

$$\text{diC}_8 \text{ PIP}_2 \text{ rescue ratio} = \frac{I_{PIP_2} - I_{PLL}}{I_{Baseline} - I_{PLL}} \tag{3}$$

where $I_{Baseline}$ is the steady state current before PLL application, $I_{PLL}$ is the steady state current during PLL treatment, and $I_{PIP2}$ is the peak current following PIP₂ application.

The normalized $G/G_{max}$–V curves were fit with a Boltzmann equation:

$$\frac{G}{G_{max}} = \frac{1}{1 + e^{\frac{-zF(V_m - V_{50})}{RT}}} \tag{4}$$

where $G/G_{max}$ is the normalized channel conductance, $z$ is the number of equivalent gating charges, $V_m$ is the membrane voltage, $V_{1/2}$ is the voltage at which $G/G_{max}$ is 0.5, F is the Faraday's constant, R is the gas constant, and $T$ is the absolute temperature.

**Molecular docking and atomistic simulations**. The cryo-EM structure of mouse TMEM16A in the $Ca^{2+}$-bound state (PDB 5OYB) was used in all molecular docking and atomistic simulations. The sequence in the cryo-EM structure has four additional residues (448EAVK451) compared with the construct used in the current experimental studies. This segment was included in all simulations to minimize any unnecessary disruption of the origin cryo-EM structure. Several short loops in cytosolic domain (T260-M266; L467-F487, L669-K682) are absent and they were rebuilt using the ProMod3 tool with Swiss-PBD server. The backbone of missing residues was first modeled with the Swiss-PDB fragment library, and the side chains were then constructed from a rotamer library[63]. The N⁻ and C-terminal segments (M1-P116 and E911-L960) as well as a long loop in the cytosolic domain (Y131-V164) are also absent in the cryo-EM structure. These regions are presumably dynamic and thus not included in simulations. The residues before and after the missing part were capped with either an acetyl group (for N-termini) or a N-methyl amide (for C-termini). To minimize the effects of missing residues on the cytosolic domain, the backbone of structured parts of the cytosolic domain (E121-E129, L165-R219, K228-L231, S243-T257, G267-L283, D452-S466, P890-R910) was harmonically restrained with a force constant of 1 kcal/(mol.Å²) during all simulations described below.

Three sets of atomistic simulations were performed, all involving $Ca^{2+}$-bound state of TMEM16A. In the first set, the full−length PIP₂ molecule was first docked into the

putative binding site identified from mutagenesis scanning using Autodock Vina[64]. Briefly, docking was carried out over a search space of $29 \times 32 \times 57$ Å covering the putative $PIP_2$ binding site formed by the intracellular portions of TMs 3, 4, 5, and TM2–3 linker. The docked structure was then used to initiate multiple atomistic simulations in explicit membrane and water to examine the stability of $PIP_2$ in the proposed binding site and atomistic details of $PIP_2$-TMEM16A interactions. The TMEM16A structure (with or without $PIP_2$) was first inserted in model POPC lipid bilayers and then solvated in TIP3P water using the CHARMM-GUI web server[65]. The second and third sets of simulations involved the $Ca^{2+}$-bound, but $PIP_2$-free state TMEM16A. In the second set, 5% of randomly chosen POPC molecules were replaced with full–length $PIP_2$ molecules, to examine spontaneous rebinding of $PIP_2$. In the third set of simulations, 20 $diC_2$ $PIP_2$ molecules (head group) were randomly placed in solution, to explore possible binding sites of $PIP_2$ head group alone. All final solvated systems were neutralized and 150 mM KCl added. The final simulation boxes contain about ~600 lipid molecules (POPC and/or $PIP_2$) and ~70,000 water molecules and other solutes, with a total of ~316,000 atoms and dimensions of ~$150 \times 150 \times 135$ Å$^3$. The CHARMM36m all-atom force field[66] and the CHARMM36 lipid force field[67] were used. $PIP_2$ parameters were adopted from a previous study[68]. All simulations were performed using CUDA-enabled Amber14[69]. Long-range electrostatic interactions were described by the Particle Mesh Ewald (PME) algorithm with a cutoff of 12 Å. Van der Waals interactions were cutoff at 12 Å with a smooth switching function starting at 10 Å. The lengths of hydrogen-containing covalent bonds were constrained using SHAKE and the MD time step was set at 2 fs. The temperature was maintained at 298 K using the Langevin dynamics with a friction coefficient of 1 ps$^{-1}$. The pressure was maintained semi-isotropically at 1 bar at both $x$ and $y$ (membrane lateral) directions using the Monte Carlo barostat method[70,71].

All systems were first minimized for 5000 steps using the steepest descent algorithm, followed by a series of equilibration steps where the positions of heavy atoms of the protein/lipid were harmonically restrained with restrained force constants gradually decreased from 10 to 0.1 kcal/(mol.Å$^2$). In the last equilibration step, only protein heavy atoms were harmonically restrained, and the system was equilibrated 10 ns in under NPT (constant particle number, pressure, and temperature) conditions. All production simulations were performed under NPT conditions. Three independent 400-ns simulations for the $PIP_2$-bound state of TMEM16A in pure POPC bilayers. For spontaneously binding simulations, four simulations of 200–700 ns were performed $PIP_2$-free TMEM16A in either mixed POPC-$PIP_2$ bilayers or in a POPC membrane with free, soluble $PIP_2$ headgroups. Snapshots were saved every 50 ps for analysis. For analysis of $PIP_2$-TMEM16A contacts, only 100–400 ns trajectories were included. Salt–bridge interactions between $PIP_2$ and basic residues in the putative TMEM16A binding site were considered formed whenever the minimum distance between any atom of the ammonium (Lys) or guanidinium (Arg) moieties and oxygen atoms in 1-/4-/5- phosphate group was no >5 Å. The pore profile of the putative ion permeation pathway was calculated using program HOLE[72]. The PMF of water was calculated directly from the histograms of water distribution along the membrane normal. The 3D water density distribution near the ion permeation pathway was calculated using a cubic grid with a resolution of 0.5 Å.

**Structure and sequence analysis**. PDB coordinate files were downloaded from the Protein Data Bank website https://www.rcsb.org/. All figures were generated in Pymol software (Schrödinger, Inc.). Sequence alignment was performed in UniProt online software (https://www.uniprot.org/align/).

**Statistical analysis**. All statistical analyses were performed in Prism software (GraphPad). Two-tailed Student's $t$-test was used for single comparisons between two groups (paired or unpaired), and one-way ANOVA was used for multiple comparisons. Comparisons yielding $p$-values < 0.05 are considered to be statistically significant. Data in summary graphs are represented as mean ± standard error of the mean (SEM), and each data point represents an independent recording experiment. Symbols *, **, ***, and **** denote statistical significance corresponding to $p$-value <0.05, <0.01, <0.001, <0.0001, respectively.

**Reporting summary**. Further information on research design is available in the Nature Research Reporting Summary linked to this article.

## Data availability

Data supporting the findings of this manuscript are available from the corresponding author upon reasonable request. A reporting summary for this article is available as a Supplementary Information file. The source data underlying Figs. 1b–d, f, g, 2, 3a, c, 5d–g, 6a–d, f, and Supplementary Figs. 1–4, 6–9a, 11–14 are provided as a Source Data file.

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

## Acknowledgements

We thank Dr L.Y. Jan (UCSF) for the initial support of this project, M. Young (Duke) for help with data analysis, Y. Sun (Duke) for help with PIP₂ docking, and Dr S-Y. Lee (Duke) for critical comments on the manuscript. This work was supported by grants NIH-R00-NS086916 (H.Y.), NIH-DP2-GM126898 (H.Y.), NIH-R01-HL142301 (J.C.), the Whitehead Foundation (H.Y.), and the American Heart Association Pre-Doctoral Fellowship (S.C.L.).

## Author contributions

H.Y. conceived and supervised the project. H.Y. and S.C.L. designed the research. S.C.L. performed all electrophysiology experiments, molecular biology, and data analysis. Z.J. performed docking and computation under the guidance of J.C. S.C.L., Z.J., J.C. and H.Y. wrote the manuscript.

## Additional information

**Competing interests:** The authors declare no competing interests.

