## [Peer Review File · Nature Communications]

Reviewers' Comments:

Reviewer #1:

Remarks to the Author:

The paper by Le et al. builds on previous published papers (Br J Pharmacol. 2017; 174(18):2984-2999. and Biochim Biophys Acta Mol Cell Biol Lipids. 2018;1863(3):299-312) that demonstrated that PIP2 is an allosteric modulator of the TMEM16A channel. This manuscript provides some new insights into the structural features that might represent a binding site for PIP2 on the channel. The authors proposed that PIP2 dissociation from a binding site in proximity of the ion permeation pathways promotes pore collapse and irreversible channel rundown. The study includes experimental and simulation data that overall appears carefully executed. However, I have reservations on how some of the data have been designed and interpreted.

1) The authors use the term "desensitisation" to describe the spontaneous decline in current amplitude that occurred upon patch excision in a solution containing a supraphysiological free Ca^{2+} concentration. I would argue that the correct term to describe this process is channel "rundown". The term desensitisation is normally used to describe a physiological decline in channel function that may occur in intact cells (e.g. desensitisation of the nicotinic acetylcholine receptor that is elicited by prolonged or repeated exposure to agonists and promotes inactivation of its ion channel). In contrast, the rundown described in the paper might represent an unphysiological consequence of patch-excision in high $[Ca^{2+}]$ solution, based on two considerations: (a) To the best of my knowledge, whole-cell TMEM16A current recorded in high intracellular $[Ca^{2+}]$ do not rundown as much as in excised patches, suggesting that patch excision is a factor that promotes rundown and (b) Intracellular $[Ca^{2+}]$ of 100 μM is well above the concentration to which the channel can be exposed in native conditions in both excitable and non-excitable cells. Even in cellular microdomains, where $[Ca^{2+}]$ can reportedly be above $\sim 10 \mu M$, this high level of Ca^{2+} is only maintained for brief periods of time (and not for several minutes like in the inside out patches presented in this paper).

Thus, in summary, the electrophysiological experiments in high $[Ca^{2+}]$ /inside-out presented in this paper are outside normal physiological conditions and the current decline should therefore be described as rundown and this point explained in the text.

Furthermore, the proposed physiological role for PIP2 regulation of the channel that the authors state in the Discussion (pg 14 mid paragraph) cannot be presented as a new proposition as it was already suggested in the original paper in which PIP2 regulation was first described.

2) Regardless of the physiological significance of the process under examination, I noted that at the recent Biophysical Society meeting, a presentation on PIP2 and TMEM16A by the Suh's lab suggested that the TMEM16A splice variant (a) was PIP2 insensitive. I wonder if the authors can comment on this potential discrepancy with their work?

The authors may also consider examining the rundown of the TMEM16B channel. The TMEM16B channel is inhibited by PIP2 (Br J Pharmacol. 2017; 174(18):2984-2999), thus the rundown in high Ca^{2+} solution should be prevented or reduced (compared to that of TMEM16A), assuming that high Ca^{2+} promotes rundown of TMEM16 channels by obstructing PIP2 action.

Indeed the observation presented in your manuscript that PIP2 antagonises the high Ca^{2+} -induced rundown of TMEM16A is not per se a demonstration that rundown is due to PIP2 dissociation from a site on the channel (i.e. it does not demonstrate a causal relationship). The paper also does not provide an explanation for why excision of patches into low Ca^{2+} solution does not promote rundown and whether that indicates that channel affinity for PIP2 is higher in low Ca^{2+} .

3) In the Introduction, the authors referred to the recent Ca^{2+} -bound TMEM16A structures, where

the pore seems to acquire a non-conductive collapse configuration. The authors speculate that this might reflect lack of PIP2 regulation, but equally other possibilities can be considered, such as loss of permeating ions that could trigger pore closure. The authors should elaborate on this possibility in the text.

4) Do mutations that enhance rundown in response to high Ca^{2+} also promote rundown in lower Ca^{2+} ? Generally these mutations should be characterised more fully including potential effects on single channel conductance. Another critical experiment would also be to study the effect of PIP2 depletion (induced for example by VSP activation) on the current mediated by the mutants with enhanced rate of rundown.

5) On pg 11 the effect of permeating ions on channel rundown is studied. Is the effect of PIP2 (or soluble analogous) altered in the presence of SCN-? The experiments should also be performed with anions with reduced permeability or permeability comparable to that of Cl^- .

6) NFA has a very complex effect on the TMEM16A channel, involving both activation and inhibition (e.g. PLoS One. 2014;9(1):e86734). Thus, the effect on channel rundown described by the author may just simply represent allosteric activation on the channel by NFA. The authors may consider repeating these experiments with a mere pore blocker (e.g. DIDS).

7) Statistical analysis. In several instances the number of experiments is only equal to 4. This impairs statistical analysis as this low number of experiments precludes determination of normality of distribution of the data. Thus, some of the experiments presented are inconclusive at this stage as they cannot be rigorously assessed statistically.

8) I suggest that the authors include in the suppl. Info. the sequence of the cDNA they used with the tracts corresponding to the TM domains, clearly labelled. This would help the reader in identifying the key residues on the primary structure as well as in the tertiary structure images already shown in some of the Figures.

Reviewer #2:

Remarks to the Author:

In the manuscript by Le et al., the authors reported that PIP2 binding to a putative binding site at the cytosolic interface of transmembrane segments (TMs) 3-5 to controls TMEM16A desensitization. Their proposed a "two-module" model and "pore collapse" mechanism have great innovative meaning. This theory will greatly assist in the analysis of the TMEM16A gating mechanism and the learning of other TMEM16 family ion channels. This is an interesting paper with some potential for further application. I have some comments and questions:

1. Almost all experiments were performed "Under saturating intracellular Ca^{2+} concentration" conditions. Please test that PIP2 also affects TMEM16A desensitization under low intracellular concentration of intracellular Ca^{2+} .

2. Why FL PIP2, diC8 PIP2, PI(3,4,5)P3 can control TMEM16A desensitization but PI(4)P enforces minimal.

3. The current generated by 100uM Ca^{2+} in Figure 2a and b is nearly three times different. Is there any statistical data indicating that this is a normal experimental phenomenon?

4. The current rescues the desensitized mTMEM16A channels by FL PIP2 in Figure 2b is only about 100 pA, and this value is not a large current. Does the current will recovered more if the starting current is larger? Or will it still be restored in proportion?

5. The author said "The sidechains of R451, R482 and K567 directly coordinate PIP2 (Fig. 4a), forming stable salt bridge interactions with PIP2", but I did not find R482 in Figure 4a.

6. TMEM16A cannot be activated after desensitization, but it does not indicate that it is "a thermodynamically stable state". Such a description needs to be modified.

7. I have some doubts about the assumption of pore collapse. Will it be through the combination of

Cl⁻ and some areas of the pore that cause some conformational changes? If this is the case, SCN⁻ can not achieve the effect of Cl⁻, and NFA can also achieve the effect described in the text.

Reviewer #3:

Remarks to the Author:

This is a very interesting manuscript on the mechanism by which prolonged calcium-dependent activation of TMEM16A results in channel desensitization or rundown. Son C. Le and colleagues propose a putative PIP₂ binding site at the cytosolic interface of several transmembrane segments of TMEM16A that plays a regulatory role in pore collapse and subsequent channel desensitization. The manuscript is well written, and the electrophysiology and mutagenesis work is elegantly performed and supported with additional computational tools and the available structural data on TMEM16A. The authors discuss their data thoroughly and propose two functionally distinct modules that regulate the open-close transition of the TMEM16A permeation pathway; the calcium binding site which mediates activation, and the PIP₂ binding site which mediates activation and desensitization. Besides, the proposed collapse of the pore based on electrophysiology and mutagenesis experiments could explain the limited pore size observed in the TMEM16A structures. The statistical analysis is properly performed, and the reproducibility of the data seems appropriate. The manuscript will be suitable for publication in Nature Communications if the authors can address the following concerns:

Major comments:

1. diC8 and full length (FL) PIP₂ are both shown to prevent desensitization of TMEM16A. However, the authors observe a different effect on the prevention of desensitization by these lipids. Including a dose-response relationship of the FL PIP₂, similar to that performed for the diC8 PIP₂ (Figure 1f), would help to better understand the difference in the behavior of these lipids. Alternatively, it would be great to know the rationale behind using 10 μM of the FL PIP₂ and whether the experiments with FL PIP₂ were performed at saturating concentrations of this lipid.
2. The effect on TMEM16A desensitization by PIP₂ lipids is specific to the position of the phosphate as shown in supplementary figure 1d. However, the concentrations of the lipids (PI(3,4,5)P₃, PI(4,5)P₂, and PI(4)P) used in this experiment are not specified in the figure legend or anywhere in the text. Please include this information in the manuscript.
3. Including the calcium potency data for the two pore-lining residues, P591A and I637F would show that these mutants are not affecting the calcium sensitivity and are being activated at saturated calcium concentrations.
4. It would be great to include in the manuscript the potential role of PIP₂ in the desensitization of the P566A and D481A mutant to show that these mutations perturb the conformation of the PIP₂ binding site and that the structural integrity of the putative PIP₂ binding site is important for TMEM16A desensitization, as suggested by the authors.
5. A conserved salt bridge between residues K579 and E564 seems to have an important role in desensitization since both alanine mutants K579A and E564A have an accelerated desensitization. An inverted double mutant K579E/E564K would be a powerful tool to confirm that indeed this salt bridge is essential for the desensitization of TMEM16A.

Minor comments:

1. The authors mention that the differences observed between the diC8 and the FL PIP₂ are due to a better incorporation of the FL PIP₂ in the cell membrane (Page 6; line 6, "likely due to stronger membrane association and slower dissociation kinetics of full length PIP₂ than the soluble short-chain diC8 PIP₂", and in line 16 "Even though full-length PIP₂ is more readily incorporated into cell membranes than soluble diC8 PIP₂"). However experimental data showing a difference in the incorporation of these lipids in the cell membrane is not provided in the manuscript. Please include some references and/or tone down these statements.
2. Including the distances between the side chains of R575, K567, K579, and R451 and their respective interacting partners in PIP₂ would help to visualize the PIP₂ binding site in

Supplementary figure 3d.

3. Labeling of the transmembrane segments in figure 4a, 3d and, 6e would help to locate the residues in the TMEM16A structure.
4. Please indicate the total number of replicas for all the experiments that are shown in the figures as representative recoding traces.

Point-by-point responses:

We thank the reviewers for your thoughtful inputs and constructive criticisms on our manuscript, which were very helpful for us to improve our manuscript.

The major concerns from the reviewers are the 1) physiologically relevant definition of desensitization phenomenon that we observed in TMEM16A; 2) the PIP₂-dependent modulation of TMEM16A channel opening under more physiological condition (sub-micromolar Ca²⁺); 3) and our proposed permeation pore collapsed model. To address these points, we have provided our additional explanations and clarifications, carried out new experiments and substantially revised our manuscript and figures to best address these concerns.

We have performed the following experiments to help us strengthen the conclusions that PIP₂ positively modulates TMEM16A under various Ca²⁺ concentrations (saturating Ca²⁺ and sub-micromolar Ca²⁺); lack of permeation has minimal contribution to rundown; and that pore collapse likely underlies TMEM16A's Ca²⁺-dependent rundown.

1. Characterizations of PIP₂ modulation of TMEM16A under physiologically relevant Ca²⁺.
2. Effects of PIP₂ depletion on TMEM16A's Ca²⁺ and voltage-dependent channel gating.
3. Rundown behaviors of TMEM16A WT and mutations of putative PIP₂ binding residues in low 500 nM Ca²⁺.
4. Mutations of putative PIP₂ binding residues in low Ca²⁺.
5. Lack of Cl⁻ experiments to demonstrate that permeation may not be required for rundown
6. The effects of SCN⁻ in modulating TMEM16A's sensitivity to PIP₂.

In addition, we have also performed additional experiments to address the Reviewer 1's concern about the limited number of biological replicates of 4 and determined the Ca²⁺ sensitivities of the two mutations P591A and I637F based on Reviewer 2's suggestion. Please see our point-to-point responses below.

Provided the additional functional data, we therefore made substantial changes to our manuscript to better illustrate the multifaceted roles of PIP₂ in modulating TMEM16A channel gating in a wide range of Ca²⁺. Based on the substantially added new functional data and especially the mechanistic understanding of TMEM16A desensitization under physiologically relevant range of sub-micromolar Ca²⁺, we therefore propose a more comprehensive mechanistic model of TMEM16A desensitization in both low and high Ca²⁺. Due to this change, we wish to change the title of the manuscript from the original "A phosphoinositide binding module controls TMEM16A desensitization" to "Molecular basis of PIP₂-dependent regulation of the Ca²⁺-activated chloride channel TMEM16A", which we hope the reviewers would agree better reflects our story.

Reviewer #1 (Remarks to the Author):

The paper by Le et al. builds on previous published papers (Br J Pharmacol. 2017;174(18):2984-2999. and Biochim Biophys Acta Mol Cell Biol Lipids. 2018;1863(3):299-312) that demonstrated that PIP2 is an allosteric modulator of the TMEM16A channel. This manuscript provides some new insights into the structural features that might represent a binding site for PIP2 on the channel. The authors proposed that PIP2 dissociation from a binding site in proximity of the ion permeation pathways promotes pore collapse and irreversible channel rundown. The study includes experimental and simulation data that overall appears carefully executed. However, I have reservations on how some of the data have been designed and interpreted.

1) The authors use the term “desensitisation” to describe the spontaneous decline in current amplitude that occurred upon patch excision in a solution containing a supraphysiological free Ca^{2+} concentration. I would argue that the correct term to describe this process is channel “rundown”. The term desensitisation is normally used to describe a physiological decline in channel function that may occur in intact cells (e.g. desensitisation of the nicotinic acetylcholine receptor that is elicited by prolonged or repeated exposure to agonists and promotes inactivation of its ion channel). In contrast, the rundown described in the paper might represent an unphysiological consequence of patch-excision in high $[\text{Ca}^{2+}]$ solution, based on two considerations: (a) To the best of my knowledge, whole-cell TMEM16A current recorded in high intracellular $[\text{Ca}^{2+}]$ do not rundown as much as in excised patches, suggesting that patch excision is a factor that promotes rundown and (b) Intracellular $[\text{Ca}^{2+}]$ of 100 μM is well above the concentration to which the channel can be exposed in native conditions in both excitable and non-excitable cells. Even in cellular microdomains, where $[\text{Ca}^{2+}]$ can reportedly be above $\sim 10\mu\text{M}$, this high level of Ca^{2+} is only maintained for brief periods of time (and not for several minutes like in the inside out patches presented in this paper).

Thus, in summary, the electrophysiological experiments in high $[\text{Ca}^{2+}]$ /inside-out presented in this paper are outside normal physiological conditions and the current decline should therefore be described as rundown and this point explained in the text.

We appreciate the great suggestions from the reviewer. We actually also encountered some difficulty deciding the most suitable and accurate terminology to best describe the phenomenon we observed.

Based on our inside-out patch recordings and the observed time-dependent current decay in saturating Ca^{2+} , classical channel “rundown” seems to be a good choice. However, TMEM16A current also showed time-dependent decay when recorded under whole-cell configuration as reported by the Hartzell group (Yu et al., 2015). Interestingly, Figure 3B of the Hartzell group’s paper shows that the whole cell TMEM16A current decays with very similar rate compared to our inside-out patch recordings under saturating Ca^{2+} , suggesting that patch configurations may not dictate this time-dependent current decay. Rather, we believe this phenomenon reflects an intrinsic property of TMEM16A channel. Majority of the whole-cell recordings were done using low

intracellular Ca^{2+} . Under low Ca^{2+} in the sub-micromolar range, TMEM16A current decay is less dramatic under both inside-out (Supplementary Fig. 6) and whole-cell conditions (Cruz-Rangel et al., 2016; De Jesus-Perez et al., 2018).

We agree with the reviewer that “rundown” is commonly used in artificial conditions to study channel biophysics (Becq, 1996); whereas “desensitization” is routinely used in describing loss of activity of receptors and channels in the prolonged presence of stimuli under more physiological conditions. As shown in the revised manuscript, under physiological calcium range, PIP_2 depletion indeed desensitizes the channel, which can be readily reversed by exogenous PIP_2 (Fig. 2). Interestingly, under this physiological condition, PIP_2 depletion-induced desensitization can be overcome by applying higher Ca^{2+} or membrane depolarization (Supplementary Fig. 3a,b). This observation led us to determine that PIP_2 depletion indeed reduces both the channel’s Ca^{2+} sensitivity (Supplementary Fig. 3c-e) and voltage sensitivity (Supplementary Fig. 3f-h).

Under saturating Ca^{2+} , prolonged application of exogenous PIP_2 can also partially re-activate the seemingly “dead” channels after prolonged opening in saturating Ca^{2+} (Supplementary Fig. 2c-e). In addition, the putative PIP_2 binding site mutations accelerate both TMEM16A current decay under both saturating Ca^{2+} (Fig. 3 and Supplementary Fig. 5) as well as physiological Ca^{2+} (Supplementary Fig. 6).

Based on these observations and the lack of rundown observed under sub-micromolar Ca^{2+} , we believe that “desensitization” could be a more meaningful term than “rundown” to describe our TMEM16A’s complex behaviors with which we hope the reviewer will agree. Nevertheless, to avoid confusing the readers, we have included the following sentence in page 4 of the revised manuscript: *“This phenomenon has been alternatively referred to as ‘rundown’, ‘desensitization’ or ‘inactivation’ in the literature; we will use desensitization and rundown interchangeably to describe the time-dependent TMEM16A current decay in this work.”*

Furthermore, the proposed physiological role for PIP_2 regulation of the channel that the authors state in the Discussion (pg 14 mid paragraph) cannot be presented as a new proposition as it was already suggested in the original paper in which PIP_2 regulation was first described.

We thank the reviewer for pointing out this repetitive proposition. We revised the corresponding section in Discussion to clearly acknowledge the previously reported propositions on the physiological implications of PIP_2 regulation on page 17 of the revised manuscript as follows: *“Our current study together with previous reports (De Jesus-Perez et al., 2018; Ta et al., 2017) strongly support the notion that PLC-induced PIP_2 hydrolysis not only leads to TMEM16A channel activation via the IP_3 signaling pathway, but concomitantly the subsequent reduction in plasma membrane PIP_2 concentration could provide a negative feedback to turn off TMEM16A.”*

2) Regardless of the physiological significance of the process under examination, I noted that at the recent Biophysical Society meeting, a presentation on PIP_2 and TMEM16A by

the Suh's lab suggested that the TMEM16A splice variant (a) was PIP₂ insensitive. I wonder if the authors can comment on this potential discrepancy with their work?

We appreciate the reviewer for pointing out the difference between our results and the Suh group's study in terms of PIP₂ effect on different TMEM16A splicing isoforms. Indeed, the Suh group presented during the BPS meeting that TMEM16A(ac) isoform is positively modulated by PIP₂, but not isoform TMEM16(a), which lacks the c splicing site of the EAVK motif in the TM2-3 loop. However, we show here that the desensitization/rundown of the TMEM16(a) isoform indeed depends on PIP₂. We believe that the differences in methodologies could in part explain for this discrepancy as detailed below:

The Suh group utilized a different approach to interrogate PIP₂ regulation of TMEM16A. By performing whole cell recordings of HEK293 cells co-expressing TMEM16A together with DrVSP, a voltage-sensitive lipid phosphatase (Hossain et al., 2008), they observed a time-dependent inhibition of channel activity albeit not completely, upon depolarization to activate DrVSP to dephosphorylate PIP₂. However, very low intracellular Ca²⁺ (100 nM) was used in their whole-cell recording. Under this condition, Ca²⁺, voltage and PIP₂ collectively regulate channel activation. A slight change in Ca²⁺ and voltage sensitivity of different isoforms or mutant channels can have profound effects on the observed PIP₂ effect on channel activation. In fact, previous studies by the Hartzell group suggested that the EAVK segment (c isoform) contributes substantially to TMEM16A's voltage and Ca²⁺-dependent activation (Xiao et al., 2011). Via deletion of this EAVK segment from the (ac) isoform, the resulting (a) isoform shows a marked reduction in its apparent Ca²⁺ and voltage sensitivity. It is thus likely that the lack of PIP₂ effect on TMEM16A(a) isoform the Suh group observed is due to its reduction in Ca²⁺ and voltage sensitivity. By contrast, we utilized inside-out patches under saturating Ca²⁺ of 100 μM. Under this condition, TMEM16A channels, regardless of its isoforms or the minor mutational changes of Ca²⁺ and voltage sensitivity, should be constitutively open, thereby eliminating the compounding effect of channel Ca²⁺ and voltage-dependence on PIP₂ effect.

In addition to saturating Ca²⁺, our inside-out recordings under low sub-micromolar Ca²⁺ (of 250 nM to 500 nM) also explicitly show the PIP₂ effect on TMEM16A(a) channel as evidenced by 1) the rapid inhibition of channel activity upon PIP₂ depletion via poly-L-lysine (Fig. 2a); 2) the rapid recovery of current via application of exogenous diC₈ PIP₂ in a dose-dependent manner (Fig. 2a-d); and 3) mutational effects of PIP₂ binding residues under low Ca²⁺ (Supplementary Fig. 6).

In summary, we believe the discrepancy with regards to whether TMEM16A(a) isoform is sensitive to PIP₂ regulation is very likely due to different experimental conditions used. Despite this discrepancy, both the Suh group and our lab identified R486-TMEM16A(ac)/R482-TMEM16A(a) as a putative PIP₂ binding residue. This consistent conclusion further supports that the c splicing site of EAVK segment is dispensable for PIP₂ regulation of TMEM16A.

The authors may also consider examining the rundown of the TMEM16B channel. The TMEM16B channel is inhibited by PIP₂ (Br J Pharmacol. 2017;174(18):2984-2999), thus

the rundown in high Ca^{2+} solution should be prevented or reduced (compared to that of TMEM16A), assuming that high Ca^{2+} promotes rundown of TMEM16 channels by obstructing PIP₂ action.

We appreciate the reviewer for pointing out this publication (Ta et al., 2017), where the authors reported that PIP₂ promotes TMEM16A activation but inhibits TMEM16B. However, it is intriguing that very minor and sometimes even negligible inhibitory effects of diC₈ PIP₂ on TMEM16B were shown in the paper. As shown in Figure 1 of the paper, a very high concentration of diC₈ PIP₂ (100 $\mu\text{g}/\text{ml}$ or 117 μM) only induced <20% of inhibition in the presence of 1 μM Ca^{2+} at +70 mV. The inhibitory effect of 100 $\mu\text{g}/\text{ml}$ diC₈ PIP₂ somehow became even smaller in Figure 2B. Very intriguingly, the amplitudes of PIP₂ inhibition appeared to be inconsistent in Figure 1B and Figure 2B. Under the identical condition (1 μM Ca^{2+} at +70 mV), the inhibitory effect of 100 $\mu\text{g}/\text{ml}$ diC₈ PIP₂ shown in Figure 2B seems much smaller than shown in Figure 1B. In their Figure 3B, 100 $\mu\text{g}/\text{ml}$ diC₈ PIP₂ showed almost no effect on TMEM16B current elicited by 0.6 μM Ca^{2+} with an I-V protocol, except very minor inhibition when membrane was depolarized beyond +100 mV. The effect of DrVSP in the whole cell recording seems slightly obvious but still very minimal (see Figure 5). However, critical control data in the absence of DrVSP expression or with the expression of a catalytically inactive DrVSP mutant were not shown to validate DrVSP's effect on TMEM16B current. Considering the extremely high concentration of diC₈ PIP₂ used and very minor inhibitory observed in Ta et al., we find it is difficult to conclude that PIP₂ indeed has an inhibitory effect on TMEM16B.

To further test this puzzle, we conducted additional experiments to characterize TMEM16B rundown/desensitization using a similar condition as in Ta et al. (inside-out patches, 1 μM Ca^{2+} and membrane depolarization to +80 mV). To avoid any potential artifacts of high concentration diC₈ PIP₂ on free Ca^{2+} concentration (1 μM), we used 20 μM diC₈ PIP₂ instead of 117 μM . To avoid potential Cl^- flux induced change of Cl^- equilibrium potential, we used brief membrane depolarization pulses instead of holding membrane constantly at depolarizing voltage. Instead of testing PIP₂-induced TMEM16B current reduction, we examined 1) if depletion of endogenous PIP₂ can affect TMEM16B current and 2) if exogenous PIP₂ can reverse it. As shown in Response Figure 1, TMEM16B channels undergo rapid desensitization upon PLL-mediated depletion of PIP₂ and that this PIP₂ depletion-induced TMEM16B desensitization can be robustly reversed, albeit partially, via exogenous application of 20 μM diC₈ PIP₂. Our PLL-mediated PIP₂ depletion experiment and exogenous PIP₂ rescue experiments thus suggest that PIP₂ positively regulates TMEM16B instead of inhibiting it. It is thus perplexing to us as to why we observed opposite effects of PIP₂ on TMEM16B from Ta et al. We speculate that the extremely high concentration of diC₈ PIP₂ (117 μM) used in Ta et al.'s experiments may partially contribute to this discrepancy. Such high concentrations of diC₈ PIP₂ may alter free Ca^{2+} concentration, considering the low Ca^{2+} used in their studies, as well as pH, which may subsequently lead to minor channel inhibition.

Despite the positive effects of PIP₂ on both TMEM16B and TMEM16A activation under low Ca^{2+} , these closely related channels do exhibit several differences. We found that 20 μM diC₈ PIP₂ is less effective and can only partially rescue the desensitized TMEM16B

channels following PLL treatment (Response Fig. 1a-c) compared to the full recovery observed in TMEM16A channels (Supplementary Fig. 3a-c). In addition, TMEM16B channels undergo much more pronounced rundown in 100 μM Ca^{2+} , and we did not observe any considerable effects by exogenous PIP_2 to attenuate TMEM16B channel rundown (Response Fig. 1d-f). These are important and intriguing biophysical properties of TMEM16B that require careful characterizations in the future.

Response Fig. 1. Functional characterizations of TMEM16B rundown. **a,b**, Representative recording of TMEM16B WT measured under 1 μM Ca^{2+} . Poly-L-lysine (PLL, 100 $\mu\text{g}/\text{ml}$) in the presence of 1 μM Ca^{2+} was applied to deplete membrane PIP_2 . 20 μM PIP_2 in the presence of 1 μM Ca^{2+} was applied to rescue channel activity. Representative current traces at different time points in **a** are shown in **b**. **c**, Quantifications of current ratios of inhibition by PLL and rescue by diC_8 PIP_2 . Two-tailed paired Student's t -test was used: p -value is 0.0005. **d,e**, Representative recordings of TMEM16B WT from inside-out patch in the presence of 100 μM Ca^{2+} (**a**) or under presence of 100 μM Ca^{2+} and 20 μM diC_8 PIP_2 (**b**). Representative current traces of recordings in **d** and **e** are shown below. **f,g**, Average normalized currents of TMEM16B in control (100 μM Ca^{2+}) and in the presence of 100 μM Ca^{2+} and 20 μM diC_8 PIP_2 (**f**) and

their half-decay times ($t_{1/2}$) (**g**). Two-tailed unpaired Student's *t*-test was used: p-value is 0.8468. Data are mean \pm s.e.m.

Indeed the observation presented in your manuscript that PIP₂ antagonises the high Ca²⁺-induced rundown of TMEM16A is not per se a demonstration that rundown is due to PIP₂ dissociation from a site on the channel (i.e. it does not demonstrate a causal relationship). The paper also does not provide an explanation for why excision of patches into low Ca²⁺ solution does not promote rundown and whether that indicates that channel affinity for PIP₂ is higher in low Ca²⁺.

We appreciate the reviewer's critical comments. We believe the following lines of evidence provide support for a causal relationship between PIP₂ and TMEM16A desensitization.

First, under saturating 100 μ M Ca²⁺, exogenous PIP₂ not only attenuates TMEM16A desensitization, but also can partially rescue TMEM16A activity after complete rundown (Fig. 1 and Supplementary Fig. 2). Second, under sub-micromolar Ca²⁺, depletion of PIP₂ by PLL rapidly desensitizes TMEM16A, which is readily restored by exogenous PIP₂ in a dose-dependent manner (Fig. 2). In addition, recent studies from other labs also show that depletion of endogenous PIP₂ by a voltage-sensitive phosphatase (VSP) can desensitize TMEM16A (De Jesus-Perez et al., 2018) (Ko et al. 2019). Third, the mutations of the putative PIP₂ binding site can accelerate TMEM16A rundown both in saturating and low sub-micromolar Ca²⁺ (Supplementary Fig. 6 and 7). Lastly, supporting our own computational studies, recent computational works reported in the recently published preprint from the Hartzell group (Yu et al., 2019) also show that PIP₂ can readily bind to TMEM16A in multiple different sites, one of which coincides with our proposed PIP₂ binding site. In our study, we show that the putative PIP₂ binding site can readily accommodate PIP₂. From these results, we believe that there is likely a causal relationship between PIP₂ binding and TMEM16A desensitization, even though other factors could certainly contribute to this gradual loss of channel activity. Future structural studies are needed to provide a more direct evidence of PIP₂ binding to establish the mechanism of PIP₂ regulation of TMEM16A.

We also thank the reviewer for raising the questions about different rundown rates and potential differential PIP₂ binding affinities under low and high concentrations of Ca²⁺. In fact, our additional experiments clearly illustrate that under physiological condition of low intracellular Ca²⁺ (sub-micromolar range), TMEM16A's channel activity is relatively stable (Supplementary Fig. 6). PIP₂ depletion by PLL results in rapid inhibition of channel activity that is reversed via exogenous diC₈ PIP₂ application in a dose-dependent manner (Fig. 2). We further demonstrate that this inhibition of TMEM16A following PIP₂ depletion is due to the reduction of the channel's Ca²⁺ sensitivity and voltage sensitivity (Supplementary Fig. 3). The estimated EC₅₀ is \sim 2 μ M, which is lower than the EC₅₀ of \sim 4 μ M in saturating Ca²⁺. This experiment suggests, as the reviewer pointed out, that PIP₂ binding to TMEM16A channels is likely state-dependent. We suggest that PIP₂ binds to TMEM16A with higher affinity when open probability is low, thus resulting in its more stable activity. Interestingly, using a different approach in which MgCl₂ was applied to

chelate membrane PIP₂, the Hartzell reported that diC₈ PIP₂, applied into the cytosolic side of the MgCl₂-treated membrane patch, also robustly rescues TMEM16A channel activity under 270 nM Ca²⁺ with a similar EC₅₀ value of 1.24 μM (Yu et al., 2019). This result thus further supports our conclusion that PIP₂ is necessary for channel activation under sub-micromolar Ca²⁺. We have included our additional characterizations of PIP₂ regulation under low Ca²⁺ (Fig. 2 and Supplementary Fig. 3) and corresponding discussions have been included in the revised manuscript on pages 7 and 8.

3) In the Introduction, the authors referred to the recent Ca²⁺-bound TMEM16A structures, where the pore seems to acquire a non-conductive collapse configuration. The authors speculate that this might reflect lack of PIP₂ regulation, but equally other possibilities can be considered, such as loss of permeating ions that could trigger pore closure. The authors should elaborate on this possibility in the text.

We thank the reviewer for pointing out an alternative mechanism to explain pore closure during TMEM16A rundown/desensitization. Based on the insights from the recent structural studies in addition to results from our additional experiments, we believe that loss of permeating ions is unlikely a major contributor to this process of pore closure.

The two structural studies of TMEM16A (Dang et al., 2017; Paulino et al., 2017) were conducted in the presence of physiological Cl⁻ concentration (~150 mM) but in the absence of PIP₂. Both the Ca²⁺-free and Ca²⁺-bound structures harbored non-conductive conformations despite presence of high concentration of Cl⁻. Because Cl⁻ ions are readily available during protein purification and in their grid conditions, we speculate that the lack of permeant ions is unlikely the reason why the Ca²⁺-bound structures adopted their non-conducting pores. In addition, the ion permeation pore of the Ca²⁺ bound structures, even though not wide enough to conduct Cl⁻ permeation, is indeed more continuous and wider than the ion pore of the Ca²⁺-free structures (Paulino et al., 2017). This observation strongly implies that the non-conductive collapsed state observed in the reported Ca²⁺-bound TMEM16A cryo-EM structures is a result of the conformational changes that take place following prolonged Ca²⁺ activation, during which the pore gradually closes and excludes permeation ions.

To further examine the effect of pore occupancy by permeation ions on TMEM16A desensitization, we mimicked a condition of losing majority of permeating ions by replacing most of intracellular Cl⁻ with the impermeant methanesulfonate (MES⁻) (Kleene, 1993) (136 mM MES⁻ and 4 mM Cl⁻). As our pipette solution contains 140 mM Cl⁻ (E_{Cl} ~ -50 mV), we held the membrane potential at -50 mV and hyperpolarized the membrane to -100 mV to force MES⁻ efflux through the pore from intracellular side. If lack of permeation ion plays an important role in TMEM16A desensitization, we should observe accelerated channel desensitization when MES⁻ is the dominant anion that is favored to flush out due to the hyperpolarized membrane potential (-50 mV and -100 mV). Contrasting this prediction, we did not observe noticeable difference in TMEM16A current decay compared to that observed in the symmetric 140 mM Cl⁻ condition (Supplementary Fig. 12). This result strongly suggests that reducing the permeant anion Cl⁻ occupancy in the

pore via replacing Cl^- with MES^- did not enhance or promote channel rundown or pore closure. This result has been included in the revised manuscript along with a brief discussion on page 14 of the revised manuscript.

4) Do mutations that enhance rundown in response to high Ca^{2+} also promote rundown in lower Ca^{2+} ? Generally these mutations should be characterised more fully including potential effects on single channel conductance. Another critical experiment would also be to study the effect of PIP_2 depletion (induced for example by VSP activation) on the current mediated by the mutants with enhanced rate of rundown.

We thank the reviewer the insightful questions. We have performed additional experiments of TMEM16A WT and mutations of putative PIP_2 binding residues in the presence of 500 nM (around their Ca^{2+} EC_{50} – see Supplementary Fig. 7) to assess their desensitization behaviors. We found that mutations of the putative PIP_2 binding residues that accelerate rundown in high Ca^{2+} also promote TMEM16A desensitization in lower, physiologically relevant range of Ca^{2+} (Supplementary Fig. 6). This enhanced rundown seen in these mutants is in stark contrast to what we observed in WT TMEM16A whose current remained relatively stable under low intracellular Ca^{2+} concentration (Supplementary Fig. 6). Therefore, the effects of PIP_2 dephosphorylation via DrVSP activation were not tested on these mutations. Interestingly, it should be noted that the Suh lab, by applying a different approach using DrVSP, also identified a key residue R486 (R482 in TMEM16A(a) isoform) which plays a critical role in PIP_2 binding, thus further corroborating our conclusion.

Accurate measurement of TMEM16A single channel conductance has been challenging and controversial in the field due to its small amplitudes. The reported single channel conductance values vary in a large range, from 3 to 9 pS (Adomaviciene et al., 2013; Burris et al., 2015; Lim et al., 2016; Yang et al., 2008) depending on the methods used and experimental setups. In the reported single channel recordings, the current traces are often very noisy and appeared to adopt a multiple sub-conductance states (Burris et al., 2015; Whitlock and Hartzell, 2016; Yang et al., 2008). Difficulty eliminating contaminating currents from endogenous channels in addition to the lack of highly specific and potent blockers makes it more challenging to obtain reliable measurement of single channel conductance. Furthermore, using noise analysis to estimate single channel conductance is not accurate enough and may not be able to distinguish mutational effects. Thus, owing to these technical difficulties, we could not reliably interrogate whether the mutations of the putative PIP_2 binding residues also perturbs the single channel conductance.

5) On pg 11 the effect of permeating ions on channel rundown is studied. Is the effect of PIP_2 (or soluble analogous) altered in the presence of SCN^- ? The experiments should also be performed with anions with reduced permeability or permeability comparable to that of Cl^- .

To address the first question, we measured diC_8 PIP_2 dose-response on attenuating TMEM16A rundown in the presence of SCN^- and saturating Ca^{2+} . As shown in Figure 6b,

PIP₂ also prevents TMEM16A rundown in a dose-dependent manner in high SCN⁻ solution (symmetric 112 mM SCN⁻/28 mM Cl⁻). The estimated PIP₂ EC₅₀ under this predominantly SCN⁻ condition is about 2.64 μM, which is more sensitive than the EC₅₀ of 3.95 μM in the presence of symmetric 140 mM Cl⁻. This result further supports our hypothesis that the more permeable anion SCN⁻ could stabilize channel open conformation to slow down rundown. As SCN⁻ is a large anion that favors dwelling in the pore (Qu and Hartzell, 2000), the stabilizing effect of SCN⁻ on the pore may allosterically enhance PIP₂ binding affinity to the putative PIP₂ binding site, thereby attenuating TMEM16A rundown. We have included this new data in the revised manuscript in Fig. 6b.

TMEM16A CaCC follows the permeability sequence of SCN⁻>I⁻>Br⁻>Cl⁻>F⁻ (Hartzell et al., 2005). As F⁻ can precipitate Ca²⁺, it is thus not suitable for our electrophysiology experiments of TMEM16A. Thus, we chose Br⁻ to address the reviewer's second question. We found that replacing the majority of Cl⁻ with Br⁻ (symmetric 112 mM Br⁻, 28 mM Cl⁻) reduced the rundown rate of TMEM16A under high Ca²⁺ (Supplementary Fig. 11). Interestingly, the current decay rates also follow the ion permeation sequence: the higher permeability of the anion, the slower rundown rate (SCN⁻ to be the slowest, followed by Br⁻ and then Cl⁻). By contrast, when the majority of Cl⁻ was replaced by the impermeant MES⁻ (Kleene, 1993), rundown of TMEM16A's currents carried out by the significantly reduced Cl⁻ was not altered (Supplementary Fig. 11). Together with the SCN⁻ experiments, this result further supports our conclusion that permeation ions can interfere with the stability of the conductive pore conformation to affect TMEM16A desensitization.

These additional results were incorporated into Fig. 6b, Supplementary Figs. 11 and 12 of the revised manuscript.

6) NFA has a very complex effect on the TMEM16A channel, involving both activation and inhibition (e.g. PLoS One. 2014;9(1):e86734). Thus, the effect on channel rundown described by the author may just simply represent allosteric activation on the channel by NFA. The authors may consider repeating these experiments with a mere pore blocker (e.g. DIDS).

We thank the reviewer for raising the potential complications of using NFA. We attempted to screen other available TMEM16A blockers including DIDS to test if we can repeat the NFA experiment (Response Fig. 2). Unfortunately, none of the blockers we tested were as efficient and reversible in inhibiting TMEM16A from the intracellular side as NFA.

- 1) DIDS is known to be a very poor CaCC blocker, especially from intracellular side (Qu and Hartzell, 2001). It is much less effective than flufenamates such as NFA (Hartzell et al., 2005). Indeed, we found that intracellular DIDS did not show obvious inhibitory effect on TMEM16A current as TMEM16A current desensitized with the same rate in the presence and absence of DIDS (Response Fig. 2a).
- 2) CaCCinh-A01 was shown to potently block TMEM16A when applied externally under whole-cell configuration (Bradley et al., 2014). We found that while this blocker at 50 μM can significantly block TMEM16A, the residual or remaining

continued to decay over time and resulted in permanent loss of channel activity (Response Fig. 2b).

- 3) Although T16Ainh-A01 was shown to effectively block TMEM16A from extracellular side under 117 nM Ca^{2+} (Bradley et al., 2014), we found that at 50 μM of T16Ainh-A01 failed to exert blocking effects TMEM16A from the intracellular side in the presence of 100 μM Ca^{2+} (Response Fig. 2c). Consistent with our observation, a recent study showed that intracellular T16Ainh-A01 greatly loses its potency to block TMEM16A when channel open probability is high (Sung et al., 2016). In fact, a small increase of Ca^{2+} concentration from 200 nM to 500 nM profoundly reduced the blocking effect of T16Ainh-A01 on TMEM16A from 70% to ~30-40% inhibition. Thus, under saturating 100 μM Ca^{2+} during our experiments, we believe that T16Ainh-A01 is likely unable to inhibit the fully open TMEM16A channels.
- 4) Similar contrast to NFA (at 100 μM), CaCCinh-A01 (50 μM), MONNA (50 μM), and Benzbromarone (100 μM) can markedly block TMEM16A from intracellular side (Response Fig. 2d-f). Unfortunately, unlike the reversibility seen in NFA (Response Fig. 2f), these blockers appeared very 'sticky' to the channel as they could hardly be washed off. Due to the irreversibility of these blockers, we were not able to utilize them in our experiment.

Nevertheless, we looked into the literature to have a better understanding of how NFA acts on CaCC. We found that NFA was indeed reported to have some stimulatory effects on both native CaCCs (Ledoux et al., 2005; Piper et al., 2002) and heterologously expressed TMEM16A (Bradley et al., 2014). However, we noticed that these characterizations were performed in whole-cell configuration in which very low Ca^{2+} was included in the pipette solution. More importantly, under 117 nM intracellular Ca^{2+} in which the channel exhibits time and voltage-dependent activation, NFA (at 30 μM) was found to inhibit TMEM16A activity but somehow reduced the channel deactivation of the remaining current during the hyperpolarization phase (Bradley et al., 2014). This is quite fascinating because one plausible way in which NFA could exert this effect on channel deactivation is via affecting the ion permeation pore, similar to a 'foot-in-the-door' mechanism.

In the paper referred by the reviewer (Ni et al., 2014), the authors in fact reported that NFA-mediated inhibition of TMEM16A is independent of channel gating. That is, NFA does not alter the channel's Ca^{2+} sensitivity. Also, the extent of blockage by NFA strongly depends on the permeant ions – the more permeable the permeating ions, the less inhibition by NFA. The authors also found that channel block by NFA of TMEM16A is affected by the occupancy of permeant ions in the pore and is independent of channel activation. These observations further support that NFA likely blocks TMEM16A via entering the ion channel pore.

Response Fig. 2. Characterizations of different TMEM16A blockers. a-f, Different known TMEM16A blockers exhibit differential effects on blocking TMEM16A from the intracellular side. Time-course recordings of TMEM16A WT showing channel activation by 100 μM Ca^{2+} and channel block by 200 μM DIDS (a), 50 μM CaCCinh-A01 (b), 50 μM T16Ainh-A01 (c), 50 μM MONNA (d), 100 μM Benzbromarone (e), and 100 μM NFA (f).

It should be noted that in our experiments with NFA, we used saturating concentrations of both Ca^{2+} of 100 μM and NFA of 300 μM and found that at NFA efficiently blocks TMEM16A currents from the intracellular side. NFA can also effectively block TMEM16A-CaCC when applied extracellularly under whole-cell configuration (Bradley et al., 2014; Liu et al., 2015; Romanenko et al., 2010). As NFA can block TMEM16A from both sides of the membrane, we believe that NFA could serve as a blocker via acting on ion permeation pore. As shown in Response Fig. 3, mutation of pore-lining residues I637 (I637F), which attenuates TMEM16A rundown, also perturbs NFA blockade. While NFA could block TMEM16A via other unknown mechanisms, these results provide additional support that NFA likely blocks TMEM16A via occupying the ion channel pore.

Because the stimulatory effects of NFA on TMEM16A were only observed under low Ca^{2+} concentrations (Bradley et al., 2014), we believe that the ability of NFA application to preserve channel activity following 5 minutes of $100\ \mu\text{M}\ \text{Ca}^{2+}/300\ \mu\text{M}\ \text{NFA}$ treatment solely reflects the blocking action of NFA itself. Most importantly, we utilized the steady state current amplitudes measured when NFA was completely washed off to quantify the residual currents. This approach helped us eliminate any potential undesired effects from NFA application.

Response Fig. 3. The Phe mutation of the pore-lining residue I637 alters NFA-mediated blockage of TMEM16A. **a,b**, Representative recordings showing current traces before (blue) and after (red) application of $100\ \mu\text{M}\ \text{NFA}$ in the presence of $100\ \mu\text{M}\ \text{Ca}^{2+}$ of TMEM16A WT (**a**) and I637F mutation (**b**). **c**, Quantifications of NFA-mediated inhibition of WT and I637F mutation. Two-tailed unpaired Student's *t*-test: *p*-value is 0.0058. **d**, Top view (from the extracellular side) of a Ca^{2+} -bound TMEM16A monomer (PDB code 5OYB) showing the location of I637 residue (colored in magenta). Bound Ca^{2+} spheres are shown in red spheres. TMs 3,4,5 are colored in pale green; TMs 6,7,8 are colored in sky blue; TMs 1,2,9,10 are colored in grey.

7) Statistical analysis. In several instances the number of experiments is only equal to 4. This impairs statistical analysis as this low number of experiments precludes determination of normality of distribution of the data. Thus, some of the experiments presented are inconclusive at this stage as they cannot be rigorously assessed statistically.

We thank the reviewer for raising this concern. We have gone back and performed additional recordings accordingly to increase their n numbers, which help us be more confident about our results and especially about statistical analyses. Additional repeats were performed for following experiments:

1. The Ca^{2+} dose-response curve of D481A shown in Fig. 5g and Supplementary Fig. 10g (previously Supplementary Fig. 7g) now has an n of 8 independent recordings. Data in these figures have been updated accordingly.
2. Recordings assessing the rundown responses of E564Q and G640A in Fig. 5e: each characterization of E564Q and G640A now both an n value of 8 independent recordings. Fig. 5e has been updated accordingly.
3. Recordings assessing the rundown response of TMEM16A WT in the presence of 112 mM SCN^- /28 mM Cl^- and control 140 mM Cl^- in Fig. 6b. Characterizations of the effects of SCN^- and Cl^- control have n values of 7 and 8, respectively, and the data in Fig. 6b have been updated accordingly.

With the additional replicates, we performed statistical analyses on these data sets and achieved similar statistical significances to draw our conclusions.

8) I suggest that the authors include in the suppl. Info. the sequence of the cDNA they used with the tracts corresponding to the TM domains, clearly labelled. This would help the reader in identifying the key residues on the primary structure as well as in the tertiary structure images already shown in some of the Figures.

We thank the reviewer for the helpful suggestion. We have included the protein sequence topology showing the locations of these key residues as suggested (Supplementary Fig. 4). We have also included an amino acid sequence alignment of mouse TMEM16A, human TMEM16A and other members of the TMEM16 protein family (Supplementary Fig. 4).

Reviewer #2 (Remarks to the Author):

In the manuscript by Le et al., the authors reported that PIP2 binding to a putative binding site at the cytosolic interface of transmembrane segments (TMs) 3-5 to controls TMEM16A desensitization. Their proposed a “two-module” model and “pore collapse” mechanism have great innovative meaning. This theory will greatly assist in the analysis of the TMEM16A gating mechanism and the learning of other TMEM16 family ion channels. This is an interesting paper with some potential for further application. I have some comments and questions:

1. Almost all experiments were performed “Under saturating intracellular Ca^{2+} concentration” conditions. Please test that PIP2 also affects TMEM16A desensitization under low intracellular concentration of intracellular Ca^{2+} .

We thank the reviewer for the insightful suggestion. As we discussed in our responses to Reviewer 1 (Response letter, pages 3 and 6), we have performed additional experiments on TMEM16A under sub-micromolar Ca^{2+} (Fig. 2 and Supplementary Figs. 3 and 6). In brief, we found that PIP_2 depletion by PLL strongly inhibits channel activity and exogenous $\text{diC}_8 \text{PIP}_2$ is sufficient to fully rescue TMEM16A channel activity in a dose-dependent manner (Fig. 2). We further demonstrated that the channel inhibition by PIP_2 depletion is due to the reduction in the channel's Ca^{2+} and voltage sensitivities (Supplementary Fig. 3). Furthermore, the mutations of the putative PIP_2 binding site residues accelerate channel rundown under sub-micromolar Ca^{2+} (Supplementary Fig. 6). Collectively, these experiments illustrate that PIP_2 also regulates TMEM16A channel gating under physiologically relevant Ca^{2+} range (~250 nM to 500 nM). The recent BioRxiv manuscript from the Hartzell group (Yu et al., 2019) and a Biophysical Society Meeting poster from the Suh group (Ko et al. Biophys J 2019; <https://doi.org/10.1016/j.bpj.2018.11.1223>) all suggest that TMEM16A activity is regulated by PIP_2 under lower sub-micromolar Ca^{2+} concentrations, thus further supporting our current findings.

2. Why FL PIP_2 , $\text{diC}_8 \text{PIP}_2$, $\text{PI}(3,4,5)\text{P}_3$ can control TMEM16A desensitization but $\text{PI}(4)\text{P}$ enforces minimal.

It has been shown that different phosphoinositide-regulated ion channels display different levels of specificity towards phosphoinositides. For example, all members of the inward rectifying (Kir) potassium channels show some dependence on $\text{PI}(4,5)\text{P}_2$. However, they display complex specificity toward other phosphoinositides. While Kir2.1 channels only bind to $\text{PI}(4,5)\text{P}_2$, Kir2.3 channels show weaker apparent affinity for $\text{PI}(4,5)\text{P}_2$ and can also bind to $\text{PI}(3,4,5)\text{P}_3$ (Du et al., 2004). In addition, Kir6.2 channels show the lowest PIP_2 affinity and can thus be activated by $\text{PI}(3,4)\text{P}_2$, $\text{PI}(4,5)\text{P}_2$, and $\text{PI}(3,4,5)\text{P}_3$ and even $\text{PI}(4)\text{P}$ at high concentrations (Fan and Makielski, 1997; Rohacs et al., 2003). Therefore, different phosphoinositide-regulated channels likely harbor different designs and conformations of their phosphoinositide binding sites, which render different channels with different phosphoinositide specificities.

In the recent BioRxiv preprint from the Hartzell group (Yu et al., 2019), the authors also reported that TMEM16A exhibits phosphoinositide specificity: whereas $\text{PI}(4,5)\text{P}_2$ and to a lesser extent $\text{PI}(3,4,5)\text{P}_3$ modulates TMEM16A channel gating, $\text{PI}(3,5)\text{P}_2$ exerts essentially no effects on TMEM16A. This result further strengthens our observation of phosphoinositide specificity in TMEM16A.

3. The current generated by 100uM Ca^{2+} in Figure 2a and b is nearly three times different. Is there any statistical data indicating that this is a normal experimental phenomenon?

We thank the reviewer for raising this point. Variation of absolute current amplitudes is a common feature of inside-out patch recording. Differences in channel expression levels, variations of glass pipette tip sizes and changes in channel properties can contribute to the observed changes in current amplitudes. In the case of wildtype TMEM16A, the amplitudes could vary from a couple hundred pA (measured at 80-mV driving force) to

several thousand pA. To avoid the variations of current amplitudes, we use normalized currents instead of absolute currents to compare different mutations in the manuscript in several occasions. Regardless of current amplitudes, our conclusions are not affected as they are based on the channel's kinetics. Nevertheless, to avoid potential confusion from the readers, we now replaced the representative current traces shown in the original Fig. 2b with a patch that had larger current amplitude in the Supplementary Fig. 2 of the revised manuscript. In addition, we have also included statistical analysis in Supplementary Fig. 2 to summarize the effects of FL PIP₂ on partially rescuing TMEM16A activity after complete desensitization.

4. The current rescues the desensitized mTMEM16A channels by FL PIP₂ in Figure 2b is only about 100 pA, and this value is not a large current. Does the current will recovered more if the starting current is larger? Or will it still be restored in proportion?

We totally agreed with the reviewer that current rescue by FL PIP₂ after complete desensitization is fairly small. In the original Fig. 2b, we showed a representative example that demonstrates the time-dependent rescue of TMEM16A current. This was a very challenging experiment because we had to repetitively perfuse FL PIP₂ and monitored the channel activity over time course of >25 minutes and time-dependent changes of current rescue also made quantification difficult. To eliminate potential misinterpretation, we have now moved Fig. 2 into Supplementary Fig. 2, which includes data from control condition (EGTA application) and FL PIP₂ (10 μM) application in addition to quantifications of the current rescue ratios after 2 fixed periods of treatments (EGTA control or FL PIP₂). As shown in the updated Supplementary Fig. 2, whereas EGTA application resulted in no detectable rescue of channel activity, application of 10 μM FL PIP₂ resulted in pronounced recovery of the channel activity. These data strongly support that TMEM16A enters an energetically stable state after prolonged opening. The percentage of FL PIP₂-induced current rescue is independent of initial current amplitudes.

5. The author said “The sidechains of R451, R482 and K567 directly coordinate PIP₂ (Fig. 4a), forming stable salt bridge interactions with PIP₂”, but I did not find R482 in Figure 4a.

We thank the reviewer for pointing out this missing information. Due to the highly flexible nature of the C-terminus of TM2-3 loop, this short peptide segment including R482 was not solved in both TMEM16A structural studies (Dang et al., 2017; Paulino et al., 2017). As R482 is located only 2 residues away from the first resolved P484 residue of TM3, we feel rather confident with the gross position of R482 in the model. Nevertheless, owing to the high flexibility of this segment, we consistently observed that the modeled-in R482 dynamically changes its sidechain orientation during our MD simulations. Considering this uncertainty, we opted to not show the position of this residue in Fig. 4a to avoid confusing the readers by over-interpreting the computational results.

Nevertheless, for better visualization of the putative PIP₂ binding site, we used a snapshot of PIP₂-bound TMEM16A structure in which the modeled-in TM2-3 loop containing R482 adopts an equilibrated conformation in Fig. 5a. In this snapshot, R482 points towards the bound PIP₂ and forms electrostatic interactions with the free phosphate of the PIP₂

headgroup. Lastly, as we discussed in our responses to Reviewer 1, the Suh group also discovered that R482 (R486 in TMEM16A(ac) isoform) may serve as a PIP₂ binding residue, which provides additional support for our proposed PIP₂ binding site.

We provided our clarifications on R482 on page 10 of the revised manuscript as: *“Different from R451 and K567, R482 resides in an unresolved short peptide in the C-terminus of TM2-3 loop preceding TM3 (Dang et al., 2017; Paulino et al., 2017). Based on our atomistic simulations, R482 has a relatively lower interaction probability of 0.60±0.10 with PIP₂, likely owing to the flexibility of this region.”*

We also added the following sentence in the legends of Fig. 4 on page 36 of the revised manuscript as: *“R482 is not shown as it is part of the highly flexible TM2-3 loop that was modeled into the structure.”*

6. TMEM16A cannot be activated after desensitization, but it does not indicate that it is “a thermodynamically stable state”. Such a description needs to be modified.

We appreciate the reviewer for raising this legitimate concern. We believe that TMEM16A channel adopts “a thermodynamically stable state” based on the following reasons.

First, our functional results strongly suggest that the desensitized channels are recalcitrant to reactivation following prolonged Ca²⁺ stimulation (Supplementary Fig. 2 in the revised manuscript). Indeed, we found that treatments of the channels with FL PIP₂ could only partially (~10%) rescue TMEM16A channel activity, and this partial rescuing process takes long time (several minutes) (Supplementary Fig. 2). This suggests that the desensitized channels, after prolonged exposure to saturating Ca²⁺, likely adopt a very stable state that can barely be reopened by exogenous PIP₂.

Second, the recent Ca²⁺-bound TMEM16A structures solved in the presence of milli-molar Ca²⁺ and in the absence of PIP₂ revealed TMEM16A’s ion permeation pore in a non-conducting state (Dang et al., 2017; Paulino et al., 2017). We strongly believe that the non-conducting state of the Ca²⁺-bound structures represents the desensitized state following application of high concentration Ca²⁺.

Taken together, these observations prompted us to speculate that TMEM16A enters this highly stable conformation that we presumed to be “thermodynamically stable state.” Since “thermodynamics” requires more careful biophysical examinations, we therefore have modified this statement according to the reviewer’s suggestion and have referred to this state as an “energetically stable state” which we referred to on page 7 of the Results section, page 16 of the Discussion section, and page 40 in the legends of Fig. 7.

7. I have some doubts about the assumption of pore collapse. Will it be through the combination of Cl⁻ and some areas of the pore that cause some conformational changes? If this is the case, SCN⁻ can not achieve the effect of Cl⁻, and NFA can also achieve the effect described in the text.

We thank the reviewer for raising the concerns about our pore collapse model of TMEM16A desensitization. Based on our intensive characterizations of TMEM16A and the available TMEM16A structures, we believe that the conformational rearrangements of the ion permeation pore of the channel play a major role in TMEM16A desensitization.

As shown in the revised Supplementary Fig. 11 in the revised manuscript, permeant ions exert differential effects on TMEM16A desensitization—the higher the permeability of the anion, the slower desensitization. As discussed in our responses to Reviewer 1's point #5 (page 8 of the response letter), we showed that diC₈ PIP₂ also prevents TMEM16A rundown in a dose-dependent manner in high SCN⁻ solution (symmetric 112 mM SCN⁻ /28 mM Cl⁻) with an estimated diC₈ PIP₂ EC₅₀ of 2.64 μM, which is more sensitive than the EC₅₀ of 3.95 μM in the presence of symmetric 140 mM Cl⁻. Similarly, we found that replacing the majority of Cl⁻ with Br⁻ (symmetric 112 mM Br⁻, 28 mM Cl⁻) reduced the rundown rate of TMEM16A under high Ca²⁺ (Supplementary Fig. 11). By contrast, when the majority of Cl⁻ was replaced by the impermeant MES⁻ (Kleene, 1993), rundown of TMEM16A's currents carried out by the significantly reduced Cl⁻ was not altered (Supplementary Fig. 11). Together with the SCN⁻ experiments, these results are suggestive of the possibility that the more permeable anions could dwell within the ion pore, hence stabilizing and preventing it from collapsing. The effects of CaCC blocker NFA and the mutations of two pore-lining residues (P591A and I637F) on attenuating TMEM16A desensitization further support this model.

In addition to our functional observations, Paulino et al. 2017 observed in the EM structure of Ca²⁺-bound TMEM16A that the ion conduction pore at the constriction site is indeed continuous and tends to approach a more conducting state when compared to the ion permeation pore in the apo, Ca²⁺-free TMEM16A structure (Paulino et al., 2017). This observation led us to speculate that the constriction site in the ion permeation pore may undergo transient conformational changes to widen during channel gating. Following prolonged Ca²⁺ stimulation, we postulate that this constriction site may gradually collapse to give rise to the non-conducting conformation observed both in the structure and our functional studies. Interestingly, recent elegant studies on the mechanisms desensitization of the Cl⁻-permeable glycine receptor (GlyR) unambiguously demonstrate that the desensitization gate constricts following prolonged glycine activation, thereby closing the conduction pore and desensitizing the channel (Du et al., 2015; Gielen et al., 2015). Future structural, functional and computational studies are thus needed to further validate the pore collapse model of TMEM16A desensitization.

Reviewer #3 (Remarks to the Author):

This is a very interesting manuscript on the mechanism by which prolonged calcium-dependent activation of TMEM16A results in channel desensitization or rundown. Son C. Le and colleagues propose a putative PIP₂ binding site at the cytosolic interface of several transmembrane segments of TMEM16A that plays a regulatory role in pore collapse and subsequent channel desensitization. The manuscript is well written, and the electrophysiology and mutagenesis work is elegantly performed and supported with additional computational tools and the available structural data on TMEM16A. The

authors discuss their data thoroughly and propose two functionally distinct modules that regulate the open-close transition of the TMEM16A permeation pathway; the calcium binding site which mediates activation, and the PIP₂ binding site which mediates activation and desensitization. Besides, the proposed collapse of the pore based on electrophysiology and mutagenesis experiments could explain the limited pore size observed in the TMEM16A structures. The statistical analysis is properly performed, and the reproducibility of the data seems appropriate. The manuscript will be suitable for publication in Nature Communications if the authors can address the following concerns:

Major comments:

1. diC₈ and full length (FL) PIP₂ are both shown to prevent desensitization of TMEM16A. However, the authors observe a different effect on the prevention of desensitization by these lipids. Including a dose-response relationship of the FL PIP₂, similar to that performed for the diC₈ PIP₂ (Figure 1f), would help to better understand the difference in the behavior of these lipids. Alternatively, it would be great to know the rationale behind using 10 μM of the FL PIP₂ and whether the experiments with FL PIP₂ were performed at saturating concentrations of this lipid.

We thank the reviewer for raising this intriguing question. First, due to the long fatty acid acyl chains, full-length (FL) PIP₂ molecules, such as that isolated from porcine brain (Avanti Polar Lipids, Cat #840046), tend to self-assemble in aqueous solutions, thereby requiring sonication on ice prior to recordings (Gamper et al., 2004; Gamper and Rohacs, 2012; Rohacs et al., 1999; Schulze et al., 2003). This sonication step results in formation of small micelles, thus facilitating fusion into the membrane patches (Flanagan et al., 1997; Rohacs et al., 1999). Also, FL PIP₂ is known to accumulate in the membrane patch and is thus difficult to control its effective concentration (Gamper and Rohacs, 2012; Rohacs et al., 2003). This property makes measurement of dose-response relationship of the FL PIP₂ difficult and is the major reason why soluble analog diC₈ PIP₂ has been commonly used to study the PIP₂ dose-dependent effects (Gamper and Rohacs, 2012; Suh and Hille, 2008).

Second, the rationale behind which we chose to use 10 μM FL PIP₂ was based on the dose-response relationship of diC₈ PIP₂ on preventing TMEM16A channel desensitization in high Ca²⁺ and rescuing TMEM16A current in low μM Ca²⁺ (Figs. 1g and 2d). diC₈ PIP₂ dose-dependently prevents TMEM16A current decay with an estimated EC₅₀ of ~4 μM and rescues TMEM16A current with an EC₅₀ of ~2 μM. Based on these estimations, we believe that 10 μM FL PIP₂ is likely close to the saturating concentration and is sufficient to prevent channel rundown.

2. The effect on TMEM16A desensitization by PIP₂ lipids is specific to the position of the phosphate as shown in supplementary figure 1d. However, the concentrations of the lipids (PI(3,4,5)P₃, PI(4,5)P₂, and PI(4)P) used in this experiment are not specified in the figure legend or anywhere in the text. Please include this information in the manuscript.

We thank the reviewer for pointing out the missing information. We used PI(3,4,5)P₃, PI(4,5)P₂, and PI(4)P all at 20 μM in the presence of 100 μM Ca²⁺. We have included this information in the legend of Supplementary Fig. 1d of the revised manuscript.

3. Including the calcium potency data for the two pore-lining residues, P591A and I637F would show that these mutants are not affecting the calcium sensitivity and are being activated at saturated calcium concentrations.

We thank the reviewer for the insightful suggestion. We tested the Ca²⁺ dependence of P591A and I637F mutant channels, both of which show enhanced Ca²⁺ sensitivity with EC₅₀ values of 0.26 ± 0.02 μM and 0.04 ± 0.001 μM, respectively (WT: 0.42 ± 0.03 μM, Supplementary Fig. 13g-j). Due to the increase of their apparent Ca²⁺ sensitivities, both mutant channels were thus fully open under 100 μM Ca²⁺.

4. It would be great to include in the manuscript the potential role of PIP₂ in the desensitization of the P566A and D481A mutant to show that these mutations perturb the conformation of the PIP₂ binding site and that the structural integrity of the putative PIP₂ binding site is important for TMEM16A desensitization, as suggested by the authors.

We thank and appreciate the reviewer for raising this insightful point. We think that these highly conserved residues likely play some roles in stabilizing the structural integrity of the channel at this putative PIP₂ binding pocket.

First, the cytoplasmic turn of TM4-TM5 is short, and Pro566 is situated at this abrupt turn between the two helices. We speculate that the presence of Pro566 provides essential rigidity to stabilize TM4-TM5. As a result, we found that P566A mutation profoundly accelerated channel rundown even though this mutation did not alter the channel's Ca²⁺ sensitivity (Fig. 5d-g). Consistent with this notion, when we introduced a glycine, which is even more flexible than alanine, the P566G mutation undergoes drastic desensitization (Response Fig. 4). We therefore posit that P566 likely plays a crucial role in the structural integrity of the proposed PIP₂ pocket.

Response Fig. 4. Mutations of P566 dramatically enhance TMEM16A desensitization. Quantifications of the half decay time ($t_{1/2}$) of TMEM16A P566A and P566G. Two-tailed unpaired Student's t -test: p -value is <0.0001 . TMEM16A WT's $t_{1/2}$ is 89.7 ± 6.0 sec.

Regarding D481, this residue is highly conserved among TMEM16 proteins (Supplementary Fig. 4). Unfortunately, the region covering D481 was not resolved in the cryo-EM structures of TMEM16A (Dang et al., 2017; Paulino et al., 2017). As D481 precedes R482 residue, which is one of the proposed PIP₂ binding residues, we speculate that disruption of D481 may perturb the local environment of the putative PIP₂ binding site.

Future experiments are needed to gain better insights into the structural integrity of the PIP₂ binding pocket in general and the roles of D481 and P566 residues in particular.

5. A conserved salt bridge between residues K579 and E564 seems to have an important role in desensitization since both alanine mutants K579A and E564A have an accelerated desensitization. An inverted double mutant K579E/E564K would be a powerful tool to confirm that indeed this salt bridge is essential for the desensitization of TMEM16A.

We thank the reviewer for the great suggestion. We have tested the inverted double mutation K579E/E564K, as well as single mutations K579E and E564K (Response Fig. 5). The charge reversal single mutations K579E and E564K showed very fast and almost instantaneous rundown upon channel activation by 100 μ M Ca²⁺. However, the double mutation K579E/E564K failed to slow down the rundown process as would be expected if the salt bridge is reformed (Response Fig. 5). Although double mutant cycles provide valuable information about the stability of salt bridges, it is not uncommon that experimental double-mutant cycles for salt bridges fail to work because this strategy only

works when there is no significant impact on the structure or stability caused by either individual mutation (Bosshard et al., 2004). The stabilizing effect of a salt bridge is not solely determined by direct Coulombic interaction between the charges; it also depends on a number of other factors such as the geometry of the salt bridge, the distance between the interacting charges, pH, the desolvation energy used to change the hydration shell of the charges and other background interactions. More systematic experimental and computational studies are needed in the future to better understand the role of this evolutionarily highly conserved salt bridge in the transmembrane domains of the TMEM16/TMEM63 super family.

Response Fig. 5. Potential roles of the highly conserved salt bridge between TM4 and TM5 segments. Functional characterizations of TMEM16A’s salt bridge mutations of E564A, E564K, K579A, K579E, and the double reverse mutations E564K/K579E. TMEM16A WT’s $t_{1/2}$ is 89.7 ± 6.0 sec.

Minor comments:

1. The authors mention that the differences observed between the diC8 and the FL PIP2 are due to a better incorporation of the FL PIP2 in the cell membrane (Page 6; line 6, “likely due to stronger membrane association and slower dissociation kinetics of full length PIP2 than the soluble short-chain diC8 PIP2”, and in line 16 “Even though full-length PIP2 is more readily incorporated into cell membranes than soluble diC8 PIP2”). However experimental data showing a difference in the incorporation of these lipids in the cell membrane is not provided in the manuscript. Please include some references and/or tone down these statements.

We agree with the reviewer that our results did not show direct experimental evidence to quantify membrane partition of diC₈ and full-length PIP₂. We have added additional references and modified our statement on page 6 of the revised manuscript to:

“This phenomenon is likely due to the different acyl chain lengths of the PIP₂ molecules. The water soluble short-chain diC₈ PIP₂ is more likely to be washed off the membrane than FL PIP₂, thus resulting in less sustained effect (Gamper and Rohacs, 2012; Rohacs et al., 1999; Rohacs et al., 2003)”.

We have also removed the statement “Even though full-length PIP₂ is more readily incorporated into cell membranes than soluble diC₈ PIP₂...” on page 6 of the original manuscript.

2. Including the distances between the side chains of R575, K567, K579, and R451 and their respective interacting partners in PIP₂ would help to visualize the PIP₂ binding site in Supplementary figure 3d.

We thank the reviewer for the suggestion. We have included the distances between the basic residues and PIP₂ headgroup in Supplementary Fig. 9d (Previously Supplementary Fig. 3d).

3. Labeling of the transmembrane segments in figure 4a, 3d and, 6e would help to locate the residues in the TMEM16A structure.

We thank the reviewer for the suggestion. We have included the labels for the TM segments in these figures.

4. Please indicate the total number of replicas for all the experiments that are shown in the figures as representative recoding traces.

We thank the reviewer for the suggestion. We have included the number of replicates in the figure legends where only representative current traces were shown in the figure panels.

References

- Adomaviciene, A., Smith, K.J., Garnett, H., and Tamaro, P. (2013). Putative pore-loops of TMEM16/anoctamin channels affect channel density in cell membranes. *J Physiol* 591, 3487-3505.
- Becq, F. (1996). Ionic channel rundown in excised membrane patches. *Biochim Biophys Acta* 1286, 53-63.
- Bosshard, H.R., Marti, D.N., and Jelesarov, I. (2004). Protein stabilization by salt bridges: concepts, experimental approaches and clarification of some misunderstandings. *J Mol Recognit* 17, 1-16.
- Bradley, E., Fedigan, S., Webb, T., Hollywood, M.A., Thornbury, K.D., McHale, N.G., and Sergeant, G.P. (2014). Pharmacological characterization of TMEM16A currents. *Channels (Austin)* 8, 308-320.
- Burris, S.K., Wang, Q., Bulley, S., Neeb, Z.P., and Jaggar, J.H. (2015). 9-Phenanthrol inhibits recombinant and arterial myocyte TMEM16A channels. *Br J Pharmacol* 172, 2459-2468.
- Cruz-Rangel, S., De Jesus-Perez, J.J., Arechiga-Figueroa, I.A., Rodriguez-Menchaca, A.A., Perez-Cornejo, P., Hartzell, H.C., and Arreola, J. (2016). Extracellular protons enable activation of the calcium-dependent chloride channel TMEM16A. *J Physiol*.
- Dang, S., Feng, S., Tien, J., Peters, C.J., Bulkley, D., Lolicato, M., Zhao, J., Zuberbuhler, K., Ye, W., Qi, L., *et al.* (2017). Cryo-EM structures of the TMEM16A calcium-activated chloride channel. *Nature* 552, 426-429.
- De Jesus-Perez, J.J., Cruz-Rangel, S., Espino-Saldana, A.E., Martinez-Torres, A., Qu, Z., Hartzell, H.C., Corral-Fernandez, N.E., Perez-Cornejo, P., and Arreola, J. (2018). Phosphatidylinositol 4,5-bisphosphate, cholesterol, and fatty acids modulate the calcium-activated chloride channel TMEM16A (ANO1). *Biochim Biophys Acta* 1863, 299-312.
- Du, J., Lu, W., Wu, S., Cheng, Y., and Gouaux, E. (2015). Glycine receptor mechanism elucidated by electron cryo-microscopy. *Nature* 526, 224-229.
- Du, X., Zhang, H., Lopes, C., Mirshahi, T., Rohacs, T., and Logothetis, D.E. (2004). Characteristic interactions with phosphatidylinositol 4,5-bisphosphate determine regulation of kir channels by diverse modulators. *J Biol Chem* 279, 37271-37281.
- Fan, Z., and Makielski, J.C. (1997). Anionic phospholipids activate ATP-sensitive potassium channels. *J Biol Chem* 272, 5388-5395.
- Flanagan, L.A., Cunningham, C.C., Chen, J., Prestwich, G.D., Kosik, K.S., and Janmey, P.A. (1997). The structure of divalent cation-induced aggregates of PIP2 and their alteration by gelsolin and tau. *Biophys J* 73, 1440-1447.
- Gamper, N., Reznikov, V., Yamada, Y., Yang, J., and Shapiro, M.S. (2004). Phosphatidylinositol [correction] 4,5-bisphosphate signals underlie receptor-specific Gq/11-mediated modulation of N-type Ca²⁺ channels. *J Neurosci* 24, 10980-10992.
- Gamper, N., and Rohacs, T. (2012). Phosphoinositide sensitivity of ion channels, a functional perspective. *Subcell Biochem* 59, 289-333.
- Gielen, M., Thomas, P., and Smart, T.G. (2015). The desensitization gate of inhibitory Cys-loop receptors. *Nat Commun* 6, 6829.
- Hartzell, C., Putzier, I., and Arreola, J. (2005). Calcium-activated chloride channels. *Annu Rev Physiol* 67, 719-758.

Hossain, M.I., Iwasaki, H., Okochi, Y., Chahine, M., Higashijima, S., Nagayama, K., and Okamura, Y. (2008). Enzyme domain affects the movement of the voltage sensor in ascidian and zebrafish voltage-sensing phosphatases. *J Biol Chem* 283, 18248-18259.

Kleene, S.J. (1993). Origin of the chloride current in olfactory transduction. *Neuron* 11, 123-132.

Ledoux, J., Greenwood, I.A., and Leblanc, N. (2005). Dynamics of Ca²⁺-dependent Cl⁻ channel modulation by niflumic acid in rabbit coronary arterial myocytes. *Mol Pharmacol* 67, 163-173.

Lim, N.K., Lam, A.K., and Dutzler, R. (2016). Independent activation of ion conduction pores in the double-barreled calcium-activated chloride channel TMEM16A. *J Gen Physiol* 148, 375-392.

Liu, Y., Zhang, H., Huang, D., Qi, J., Xu, J., Gao, H., Du, X., Gamper, N., and Zhang, H. (2015). Characterization of the effects of Cl⁻ channel modulators on TMEM16A and bestrophin-1 Ca²⁺(+) activated Cl⁻ channels. *Pflugers Arch* 467, 1417-1430.

Ni, Y.L., Kuan, A.S., and Chen, T.Y. (2014). Activation and inhibition of TMEM16A calcium-activated chloride channels. *PLoS One* 9, e86734.

Paulino, C., Kalienkova, V., Lam, A.K.M., Neldner, Y., and Dutzler, R. (2017). Activation mechanism of the calcium-activated chloride channel TMEM16A revealed by cryo-EM. *Nature* 552, 421-425.

Piper, A.S., Greenwood, I.A., and Large, W.A. (2002). Dual effect of blocking agents on Ca²⁺-activated Cl⁻ currents in rabbit pulmonary artery smooth muscle cells. *J Physiol* 539, 119-131.

Qu, Z., and Hartzell, H.C. (2000). Anion permeation in Ca²⁺-activated Cl⁻ channels. *J Gen Physiol* 116, 825-844.

Qu, Z., and Hartzell, H.C. (2001). Functional geometry of the permeation pathway of Ca²⁺-activated Cl⁻ channels inferred from analysis of voltage-dependent block. *J Biol Chem* 276, 18423-18429.

Rohacs, T., Chen, J., Prestwich, G.D., and Logothetis, D.E. (1999). Distinct specificities of inwardly rectifying K⁺ channels for phosphoinositides. *J Biol Chem* 274, 36065-36072.

Rohacs, T., Lopes, C.M., Jin, T., Ramdya, P.P., Molnar, Z., and Logothetis, D.E. (2003). Specificity of activation by phosphoinositides determines lipid regulation of Kir channels. *Proc Natl Acad Sci U S A* 100, 745-750.

Romanenko, V.G., Catalan, M.A., Brown, D.A., Putzier, I., Hartzell, H.C., Marmorstein, A.D., Gonzalez-Begne, M., Rock, J.R., Harfe, B.D., and Melvin, J.E. (2010). Tmem16A encodes the Ca²⁺-activated Cl⁻ channel in mouse submandibular salivary gland acinar cells. *J Biol Chem* 285, 12990-13001.

Schulze, D., Krauter, T., Fritzenschaft, H., Soom, M., and Baukrowitz, T. (2003). Phosphatidylinositol 4,5-bisphosphate (PIP₂) modulation of ATP and pH sensitivity in Kir channels. A tale of an active and a silent PIP₂ site in the N terminus. *J Biol Chem* 278, 10500-10505.

Suh, B.C., and Hille, B. (2008). PIP₂ is a necessary cofactor for ion channel function: how and why? *Annu Rev Biophys* 37, 175-195.

Sung, T.S., O'Driscoll, K., Zheng, H., Yapp, N.J., Leblanc, N., Koh, S.D., and Sanders, K.M. (2016). Influence of intracellular Ca²⁺ and alternative splicing on the pharmacological profile of ANO1 channels. *Am J Physiol Cell Physiol* 311, C437-451.

Ta, C.M., Acheson, K.E., Rorsman, N.J.G., Jongkind, R.C., and Tammaro, P. (2017). Contrasting effects of phosphatidylinositol 4,5-bisphosphate on cloned TMEM16A and TMEM16B channels. *Br J Pharmacol* 174, 2984-2999.

Whitlock, J.M., and Hartzell, H.C. (2016). A Pore Idea: the ion conduction pathway of TMEM16/ANO proteins is composed partly of lipid. *Pflugers Arch* 468, 455-473.

Xiao, Q., Yu, K., Perez-Cornejo, P., Cui, Y., Arreola, J., and Hartzell, H.C. (2011). Voltage- and calcium-dependent gating of TMEM16A/Ano1 chloride channels are physically coupled by the first intracellular loop. *Proc Natl Acad Sci U S A* 108, 8891-8896.

Yang, Y.D., Cho, H., Koo, J.Y., Tak, M.H., Cho, Y., Shim, W.S., Park, S.P., Lee, J., Lee, B., Kim, B.M., *et al.* (2008). TMEM16A confers receptor-activated calcium-dependent chloride conductance. *Nature* 455, 1210-1215.

Yu, K., Jiang, T., Cui, Y., Tajkhorshid, E., and Hartzell, H.C. (2019). A Network of Phosphatidylinositol 4,5-bisphosphate Binding Sites Regulate Gating of the Ca²⁺-activated Cl⁻ Channel ANO1 (TMEM16A). *BioRxiv*.

Yu, K., Whitlock, J.M., Lee, K., Ortlund, E.A., Cui, Y.Y., and Hartzell, H.C. (2015). Identification of a lipid scrambling domain in ANO6/TMEM16F. *Elife* 4, e06901.

Reviewers' Comments:

Reviewer #1:

Remarks to the Author:

Thank you for your careful consideration of most of the comments that were raised. As a result of this revision the manuscript has improved substantially.

I would like to offer additional feedback on some of the experiments now presented in Fig.2 and Suppl. Fig.3.

1) PLL has notoriously non-specific effects on ion channels which may add to those due to PIP2 depletion. Indeed, the data reported in Fig.2 show a very fast and ~complete inhibition of the current; this may be indicative of direct channel inhibition by PLL independent of (or in addition to) inhibition due to PIP2 depletion. Please also note that published reports showed that prolonged DrVSP activation only partially inhibited TMEM16A currents in spite of the fact that DrVSP activation can lead to very significant removal of plasmalemmal PIP2. This notion also raises the question as to whether PLL perturbs (at least in part) TMEM16A channel function in a PIP2-independent manner.

Therefore, the interpretation of the PLL data presented in the current manuscript is significantly impaired by not knowing the specificity of PLL effect. I suggest that this is tested.

2) The assessment of "voltage sensitivity" presented in Suppl. Fig.3 appear not to be fully adequate: the data were arbitrarily normalised for the value of the conductance at +120 mV. Your own data and published work (for example Proc Natl Acad Sci U S A. 2011 108(21):8891-6) show that the maximal conductance is clearly not reached at this voltage (when $[Ca^{2+}]_i$ is low). Voltage activation curves should therefore be obtained (from adequate Boltzmann fit of instantaneous tail currents vs V_m relationships) in a broad range of $[Ca^{2+}]_i$ measured in individual patches during inside-out recordings so that the actual maximal conductance is assessed in each patch. Furthermore, the data obtained in the presence of PLL should be normalised for the maximal conductance measured in the presence, and not in the absence, of PLL. These analyses would provide an objectively more reliable assessment of the voltage sensitivity of the channel.

Reviewer #2:

None

Reviewer #3:

Remarks to the Author:

The authors have done an excellent job of revising the manuscript and they have considered and addressed all of my concerns. I have no further comments and no issues with the authors changing the title of the manuscript.

The manuscript is now suitable for publication at Nature Communications.

Point-to-point responses:

We thank the Reviewer 1 again for your thoughtful inputs and constructive criticisms on our manuscript with regards to the newly added data presented in Fig. 2 and Supplementary Fig. 3.

The major concerns from include 1) the potential off-target effects of PLL on TMEM16A's channel activity and 2) the arbitrary normalizations of the tail currents in the absence of maximal channel conductance to construct the G-V curves. We have provided our explanations as well as performed additional experiments to address these concerns. Please see our point-to-point responses below.

Reviewers' comments:

Reviewer #1 (Remarks to the Author):

Thank you for your careful consideration of most of the comments that were raised. As a result of this revision the manuscript has improved substantially.

I would like to offer additional feedback on some of the experiments now presented in Fig.2 and Suppl. Fig.3.

1) PLL has notoriously non-specific effects on ion channels which may add to those due to PIP2 depletion. Indeed, the data reported in Fig.2 show a very fast and ~complete inhibition of the current; this may be indicative of direct channel inhibition by PLL independent of (or in addition to) inhibition due to PIP2 depletion. Please also note that published reports showed that prolonged DrVSP activation only partially inhibited TMEM16A currents in spite of the fact that DrVSP activation can lead to very significant removal of plasmalemmal PIP2. This notion also raises the question as to whether PLL perturbs (at least in part) TMEM16A channel function in a PIP2-independent manner.

Therefore, the interpretation of the PLL data presented in the current manuscript is significantly impaired by not knowing the specificity of PLL effect. I suggest that this is tested.

We sincerely thank the reviewer for this critical comment on our newly added data in Fig. 2. With regards to the apparently instantaneous inhibition of TMEM16A by PLL (Fig. 2a), we believe that the reviewer might have missed the fact that PLL was applied in zero Ca^{2+} (5 mM EGTA). This rapid loss of channel activity was merely due to the withdrawal of Ca^{2+} from the perfusing solution. This is the reason why it appears that PLL application gave rise to the immediate loss of channel activity.

We also performed an additional set of experiments in which PLL was co-applied together with low sub-micromolar Ca^{2+} (0.387 μM) and found that it took 10-15 seconds before the PLL-mediated current inhibition reached a steady state (Response Fig. 1). This observation is consistent with PIP_2 being gradually depleted by PLL, which in turn leads

to TMEM16A channel inhibition, largely owing to the loss of membrane PIP₂. However, unlike the experiment shown in Fig. 2a, one caveat of this approach is that PLL was co-applied together with Ca²⁺, and thus any undesired effects of PLL on channel activity could potentially confound our interpretation. Therefore, we believe the PLL application paradigm in the absence of Ca²⁺ shown in Fig. 2a has an advantage to obviate the potential off-target effects of PLL during Ca²⁺-dependent activation. Even though we cannot exclude the possibility that PLL could remain stuck to the channel/membrane patches following its washout to alter channel activity, we believe that the effects of channel inhibition could be mostly attributed to the loss of membrane PIP₂ because: first, exogenous PIP₂ robustly recovered the channel activity (Fig. 2a-c); and second, diC₈ PIP₂ dose-dependently rescued the channel activity following this PLL treatment (Fig. 2d).

Response Fig. 1. The effect of PLL treatment on TMEM16A's channel activity under sub-micromolar Ca²⁺. **a**, 0.387 μM Ca²⁺ was applied to elicit channel opening and PLL (100 $\mu\text{g}/\text{ml}$) together with 0.387 μM Ca²⁺ were applied to deplete membrane PIP₂ and induce channel inhibition. TMEM16A channel activity was recovered by exogenous application of diC₈ PIP₂ (20 μM in the presence of 0.387 μM Ca²⁺). **b**, Current inhibition ratio during PLL treatment and current rescue ratio by exogenous diC₈ PIP₂. Two-tailed unpaired Student's t-test: p-value is <0.0001.

Furthermore, we have also performed an additional set of experiments in which TMEM16A WT was co-expressed together with the voltage-sensitive phosphatase (VSP) from the *Ciona intestinalis* (CiVSP) to test how PIP₂ dephosphorylation could alter TMEM16A's Ca²⁺-dependent activation. We reasoned that PIP₂ dephosphorylation by CiVSP could potentially mimic the effect of PLL treatment in reducing the channel's Ca²⁺ dose response. We found that co-expression of the WT CiVSP, but not the catalytically inactive CiVSP C363S mutant, reduced TMEM16A's Ca²⁺ sensitivity albeit its mild effect (Response Fig. 2). Similar to PLL treatment, the effect of PIP₂ dephosphorylation by CiVSP was only observed under the physiologically relevant range of sub-micromolar Ca²⁺ (Response Fig. 2a,b). We speculate that the mild reduction in TMEM16A's Ca²⁺ sensitivity via PIP₂ dephosphorylation by CiVSP could be explained by the incomplete dephosphorylation of membrane PIP₂ owing to the low expression level of CiVSP. This result also corroborates the observation that reviewer alluded to regarding the mild effect on TMEM16A by DrVSP.

Response Fig. 2. The effect of CiVSP-mediated PIP₂ dephosphorylation on TMEM16A's Ca²⁺-dependent activation. **a**, Representative showing Ca²⁺ concentration-dependent activation of TMEM16A when co-expressed with the catalytically inactive CiVSP C363S mutant (top) or CiVSP WT (bottom). **b,c**, Ca²⁺ dose response curves of TMEM16A co-expressed with CiVSP C363S and CiVSP WT (**b**) and their Ca²⁺ EC₅₀ values (**c**). Two-tailed unpaired Student's *t*-test: *p*-values are 0.0042 for 0.255 μM Ca²⁺ and 0.0018 for 0.387 μM Ca²⁺ in **b**; *p*-value is 0.0065 in **c**. Data are mean ± s.e.m.

2) The assessment of “voltage sensitivity” presented in Suppl. Fig.3 appear not to be fully adequate: the data were arbitrarily normalised for the value of the conductance at +120 mV. Your own data and published work (for example Proc Natl Acad Sci U S A. 2011 108(21):8891-6) show that the maximal conductance is clearly not reached at this voltage (when [Ca²⁺]_i is low). Voltage activation curves should therefore be obtained (from adequate Boltzmann fit of instantaneous tail currents vs V_m relationships) in a broad range of [Ca²⁺]_i measured in individual patches during inside-out recordings so that the actual maximal conductance is assessed in each patch. Furthermore, the data obtained in the presence of PLL should be normalised for the maximal conductance measured in the presence, and not in the absence, of PLL. These analyses would provide an objectively more reliable assessment of the voltage sensitivity of the channel.

We appreciate the reviewer for this very constructive input on this newly added piece of data. In this set of experiments, we aimed to demonstrate that PIP₂ depletion via PLL treatment right-shifts the G-V curve to provide some evidence that loss of PIP₂ could also impair the voltage-dependent activation of TMEM16A. We first activated the channel using low sub-micromolar Ca²⁺ (0.255 μM) during which the channel exhibits voltage-dependent and time-dependent activation. Holding the membrane at -60 mV, we varied

the voltage steps from -120 mV to +120 mV to obtain an I-V recording. We then briefly treated the same patch with PLL for 15-20 seconds to deplete PIP₂ then reapplying Ca²⁺ in the absence of PLL to obtain the second I-V recording. As the reviewer pointed out, we did not obtain the maximal channel conductance under this condition. Because of this limitation, we used the peak tail current measured at -60 mV following the peak depolarization step of +120 mV to construct the G-V curve. Similarly, we measured the tail currents of the PLL-treated recording from the same patch and normalized to the peak tail current from the non-treated recording. We agree with the reviewer that this is not the most accurate way to construct G-V relation curves.

Taking the reviewer's recommendation, we first measured the maximal TMEM16A current (G_{\max}) of the channel by rapidly and briefly exposing the patches to saturating 100 μM Ca²⁺ for 3-4 seconds to avoid loss of channel activity, and then quickly switched to sub-micromolar Ca²⁺ concentrations of 0.255 μM and 0.387 μM Ca²⁺ to monitor the voltage-dependent development of tail currents at -60 mV (revised Supplementary Fig. 4b). The G/G_{\max} -V curves with and without PLL pre-treatment are now presented in the revised Supplementary Fig. 4c,d. Consistent with our conclusion based on our previous quantification, the new quantification method again reveals the reduction in the apparent voltage sensitivity of TMEM16A channels under sub-micromolar Ca²⁺ after PIP₂ depletion.

Due to these new data, we think that it will be clearer to split the PIP₂ depletion effects on TMEM16A's Ca²⁺ and voltage dependence (previously Supplementary Fig. 3) into two separate figures (Supplementary Fig. 3 for Ca²⁺ dependence and Supplementary Fig. 4 for voltage dependence).

Reviewer #3 (Remarks to the Author):

The authors have done an excellent job of revising the manuscript and they have considered and addressed all of my concerns. I have no further comments and no issues with the authors changing the title of the manuscript.

The manuscript is now suitable for publication at Nature Communications.

We thank the reviewer for the kind words and enthusiasm in our work.

Reviewers' Comments:

Reviewer #1:

Remarks to the Author:

Thank you for your careful consideration of my suggestions. I believe the work now represents a very valuable contribution to the field.